# Analyzing single-cell bisulfite sequencing data with MethSCAn

Lukas P. M. Kremer [1,2] ✉, Martina M. Braun [2], Svetlana Ovchinnikova[1], Leonie Küchenhoff [1,2], Santiago Cerrizuela [2], Ana Martin-Villalba [2] ✉ & Simon Anders [1] ✉

Single-cell bisulfite sequencing (scBS) is a technique that enables the assessment of DNA methylation at single-base pair and single-cell resolution. The analysis of large datasets obtained from scBS requires preprocessing to reduce the data size, improve the signal-to-noise ratio and provide interpretability. Typically, this is achieved by dividing the genome into large tiles and averaging the methylation signals within each tile. Here we demonstrate that this coarse-graining approach can lead to signal dilution. We propose improved strategies to identify more informative regions for methylation quantification and a more accurate quantitation method than simple averaging. Our approach enables better discrimination of cell types and other features of interest and reduces the need for large numbers of cells. We also present an approach to detect differentially methylated regions between groups of cells and demonstrate its ability to identify biologically meaningful regions that are associated with genes involved in the core functions of specific cell types. Finally, we present the software tool MethSCAn for scBS data analysis (https://anders-biostat.github.io/MethSCAn).

Sequencing-based assays with single-cell resolution have offered new means to understand the differences between the cells making up a sample. Single-cell RNA sequencing (scRNA-seq) techniques have matured at great pace in recent years, with well-developed analysis methodologies, and methods to study epigenetics at single-cell resolution are rapidly catching up.

Briefly, in a bisulfite sequencing assay, DNA is treated with bisulfite, which converts unmethylated cytosines to uracils that are read as thymine in subsequent PCR, while methylated cytosines are protected from conversion. After sequencing, these conversions allow for the determination of the methylation status of all cytosines covered by reads[1]. Bisulfite sequencing can also be performed at single-cell resolution[2] and even in parallel with scRNA-seq[3–5].

In this Article, we discuss strategies to analyze single-cell bisulfite sequencing (scBS) data. We suggest improvements to current approaches, and demonstrate their value in benchmarks, using four real-world datasets. Furthermore, we discuss how to perform comparative analyses. Finally, we present MethSCAn, a comprehensive software toolkit to perform scBS data analysis.

The standard approach to analyze scBS data is based on methodology developed for the analysis of scRNA-seq data. Therefore, we start by briefly reviewing how scRNA-seq data are commonly analyzed, before we discuss scBS data analysis.

The starting point in most scRNA-seq analyses is a matrix of unique molecular identifier (UMI) counts (that is, counts of distinct RNA molecules), with one row for each cell and one column for each gene. A first goal is usually to assign cell types or states to cells. To this end, one needs to establish which cells are similar to each other, that is, quantify the distance (that is, dissimilarity) between any two given cells' transcriptional profile. A standard approach, used with minor variation in virtually all recent research and automated by popular software such as Seurat[6] or Scanpy[7], is as follows: one first accounts

[1]BioQuant Centre, University of Heidelberg, Heidelberg, Germany. [2]Division of Molecular Neurobiology, German Cancer Research Center, Heidelberg, Germany. ✉e-mail: l.kremer@dkfz-heidelberg.de; a.martin-villalba@dkfz-heidelberg.de; simon.anders@bioquant.uni-heidelberg.de

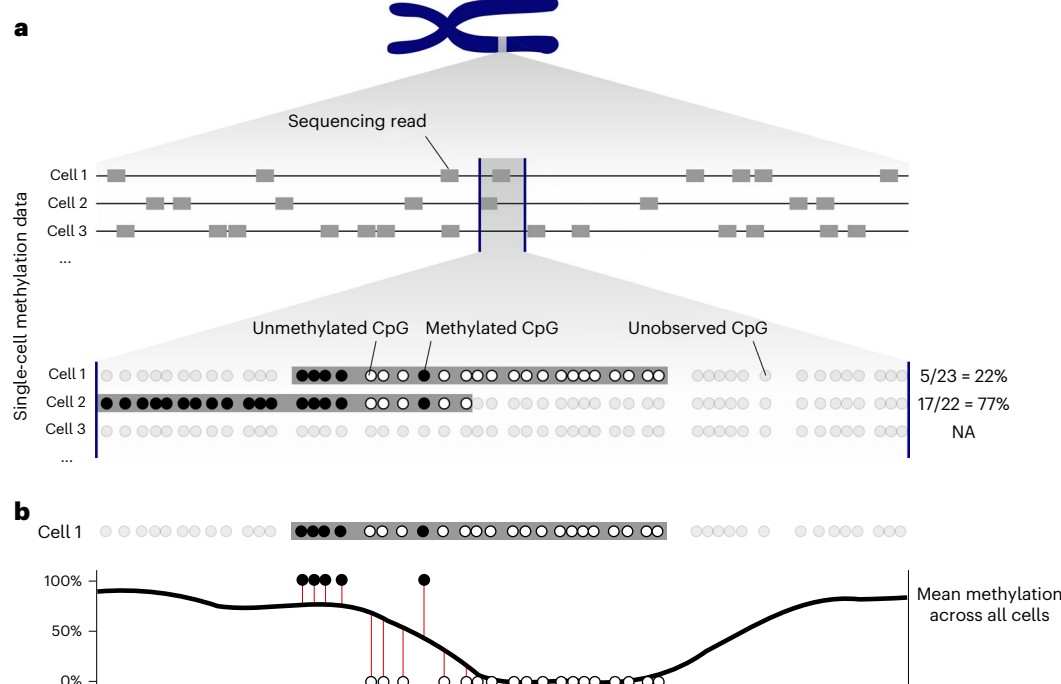

**Fig. 1 | Improved quantification of DNA methylation in a given genomic interval. a**, A genomic interval (between the vertical blue lines) along a chromosome, for which DNA methylation is to be quantified. **b**, By averaging each CpG site's methylation over all cells and subsequent smoothing, the 'average methylation' (thick black curve) is obtained. To quantify the methylation

of cell 1 from **a** relative to this average over all cells, we propose to use the cell's residuals to the smoothed curve (the lengths of the vertical red lines) and take their average, counting residuals of methylated CpGs as positive and residuals of unmethylated CpGs as negative.

for cell-to-cell variation in sequencing depth by dividing each UMI count by the respective cell's total UMI count, then transforms to a homoskedastic scale by taking the logarithm. To avoid matrix elements with zero count to be transformed to minus infinity, one typically adds a very small 'pseudocount' (often $10^{-4}$) to the normalized fractions before taking the logarithm. Now, one could use Euclidean distances of these vectors of logarithmized fractions as the dissimilarity score. However, these scores would be exceedingly noisy owing to the strong Poisson noise introduced by the many genes with very low counts. Therefore, one performs a principal component analysis (PCA), keeping only the top few (typically, 20–50) components. As Poisson noise is uncorrelated between genes, it will average out in the top principal components, as these are all linear combinations with weight on a large number of genes. Therefore, Euclidean distances between these 'PCA space' vectors provide a robust dissimilarity score. Hence, the PCA space representation is suitable as input to methods such as *t*-distributed stochastic neighbor embedding (t-SNE) and uniform manifold approximation and projection (UMAP), which provide a two-dimensional representation of the data, or to methods for clustering (assigning cells to groups by similarity) and trajectory finding (identifying elongated manifolds in PCA space and assigning cells to quasi-one-dimensional positions along them).

This procedure is commonly adapted when working with single-cell DNA methylation data, because once one gets to the PCA step, one can then continue with the established methods just mentioned. However, constructing a matrix suitable as input for PCA from methylation data requires deviation from the standard scRNA-seq workflow owing to considerable differences in data structure. First, while scRNA-seq quantifies the RNA abundance of genes or transcripts, scBS is genome-wide and thus lacks a natural choice for features in which methylation is to be quantified. Second, instead of counts, scBS generates binary data that inform us whether certain cytosines in a given cell are methylated. A simple and common approach to construct

a methylation matrix suitable for PCA, used for instance by Luo et al.[8], is to divide the genome into tiles of, for example, 100 kb size, and calculate for each cell the average methylation of the DNA within each tile. To this end, one identifies in the tile all CpG sites that are covered by at least one read and averages their methylation state, that is, one denotes as average DNA methylation of the tile in a given cell the proportion of the observed CpG sites in the tile that were found to be methylated (Fig. 1a). This yields a matrix, with one row for each cell and one column for each genomic tile, comprising numbers ('methylation fractions') between 0 and 1. This matrix is now subjected to PCA. After PCA, one can proceed with dimensionality reduction and clustering approaches known from scRNA-seq.

While this simple procedure is straightforward and produces usable results, it is not optimal. In this paper, we discuss weaknesses of the simple approach and suggest several refinements to overcome them. Using benchmarks and an application to real data, we show that our improvements substantially increase the information content of the processed data. In the main text, we explain the proposed methods and their motivation in a qualitative manner, while mathematical details are given later in Methods. Then, we demonstrate the value of our methods using benchmarks and an application to real data. Finally, we describe and demonstrate an approach to detect differentially methylated regions (DMRs) in scBS data. We also describe our software toolkit, MethSCAn, that facilitates all these analyses.

## Results

### Read-position-aware quantitation

We first discuss the task of quantifying the level of methylation in a given, fixed genomic interval. Typically, the read coverage per cell is sparse in scBS data. In the example shown in Fig. 1a, the depicted interval is covered by a single read for two of the three cells shown and no read in the third. The read from cell 2 shows much more methylation than the read from cell 1, and the standard analysis would therefore

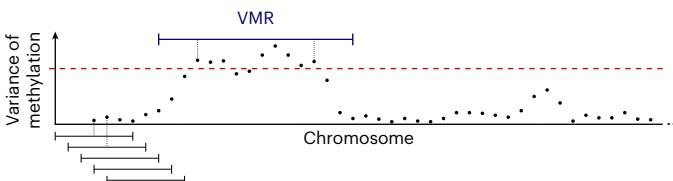

**Fig. 2 | Finding VMRs.** The chromosomes are divided up into overlapping windows (the first five are shown at the bottom), and for each window, the cells' methylation values are calculated as described and as depicted in Fig. 1b. Then, the variance of these values is calculated (where each point represents one of the overlapping windows) and a threshold (dashed red line) is chosen such that a chosen quantile of windows have a variance exceeding the threshold. Windows with above-threshold variance are merged if they overlap, yielding the VMRs.

consider cell 2 to have higher methylation in the interval than cell 1. However, given that the two reads agree wherever they overlap, a more parsimonious interpretation would be that the cells do not show a difference in methylation within the interval. Rather, both cells, and similarly maybe most other cells, might have stronger methylation in the left third of the interval than in the middle one.

Therefore, we propose to first obtain, for each CpG position, a smoothed average of the methylation across all cells and then quantify each cell's deviation from this average. In Fig. 1b, the curved line depicts such an average over all cells, and the red vertical lines show an individual cell's deviation from the ensemble average. We take the lengths of the red lines as signed values ('residuals'), being positive for lines extending upward from the curve (methylated CpG) and negative for lines extending downward (unmethylated CpG). For each cell, we then take the average over the residuals for all the CpGs in the interval that are covered by reads from this cell. In this average, we perform shrinkage toward zero via a pseudocount (to trade bias for variance, see Methods for detail) to dampen the signal in cells with low coverage of the interval.

The average thus obtained, that is, the shrunken mean of the residuals, is what we use to quantify the cell's (relative) methylation in the interval. For a genome tiled into such intervals, we thus obtain a matrix, with one row per cell and one column per interval, that can be used for downstream analysis, for example, as input for PCA. The signal-to-noise ratio in this matrix will be better than in a matrix obtained by simply averaging absolute methylation (0 or 1) over all the cells' covered CpG sites in a region. The reason for this is that we reduce the variation in situations such as the one depicted in Fig. 1a, where the methylation of the reads might differ strongly even though there is no actual evidence for a difference between the two cells.

How should one obtain the ensemble average (Fig. 1b, curved line)? A simple approach to get a value for a specific CpG would be to take all cells with read coverage for the CpG and use the fraction of these that show the CpG as methylated. However, especially when only few cells offer coverage, these averages will be very noisy. Therefore, we propose to smoothen using a kernel smoother, that is, by performing a kernel-weighted average over the CpG site's neighborhood. The kernel bandwidth (that is, the size of the neighborhood to average over) is a tuning parameter; for the examples presented here, we used 1,000 bp.

A minor remaining issue is how to deal with the case that a cell has no reads at all within a given interval. Here, it is justified to simply put zero into the matrix element, because a shrunken residual average of zero indicates that there is no evidence of the cell deviating from the mean. We slightly refine this by using an iterative imputation within the PCA ('iterative PCA', see Methods for details).

Taken together, this shrunken mean of residuals quantitation reduces the variance in comparison to simple averaging of raw methylation calls. We show further below how this improves the results.

## Finding variably methylated regions

Typically, some regions in a chromosome will have a very similar methylation status in all cells, while other regions show variability in methylation across cells. For instance, it has long been known that CpG-rich promoters of housekeeping genes are unmethylated, and that a large proportion of the remaining genome is highly methylated regardless of cell type[9]. In contrast, DNA methylation at certain genomic features such as enhancers is more dynamic[10] and thus more variable across cells. Only the latter regions are of value for our goal of quantitating the dissimilarity between cells. We call these the variably methylated regions (VMRs).

In the standard approach, one divides up (tiles) each chromosome into non-overlapping, equally sized intervals and quantitates the methylation of each tile. Such rigid placement of interval boundaries is unlikely to be optimal: for example, a VMR might be much smaller than a tile, and the signal from its CpG sites will hence be drowned out by the larger number of uninformative CpG sites that are equal in all cells, when averaging over all the CpG sites in the tile.

Therefore, we propose the following approach (Fig. 2): Divide up the chromosome into many overlapping windows that start at regular multiples of a fixed, small step size. Quantify the methylation of each cell in each window by averaging the cell's methylation residuals over all CpGs in the window, as described earlier and depicted in Fig. 1b. Next, calculate for each window the variance of these values over all cells. Select, say, the top 2% windows with the highest variances and mark them as VMRs. Wherever the marked windows overlap, merge them into one larger VMR. Then, calculate for each of these merged VMRs the methylation signal, as before, by averaging for each cell over the residuals of all contained CpG sites.

In this manner, we obtain a methylation matrix, with one row per cell and one column per VMR, that is (in a sense) richer in information and has better signal-to-noise ratio than the matrix obtained by the simple analysis sketched at the very beginning. As we demonstrate below, a PCA performed on such a matrix provided a distance metric for the cells that contains more information on biological detail than one from a simpler analysis.

## Application and benchmarks

To demonstrate the value of our proposed improvements, we benchmarked various combinations of analysis methods on five diverse single-cell methylome datasets, starting with a dataset from our own research.

**Correlating VMR methylation with gene expression.** Our dataset[11] comprises the single-cell methylomes of 1,566 cells isolated from mouse forebrains as well as matched single-cell transcriptomes of the same cells. Among these cells are distinct cell types such as oligodendrocytes, oligodendrocyte precursor cells and endothelial cells, as well as cellular substates that are part of the continuous neural stem cell differentiation trajectory. To assess whether our VMR detection method captures genomic intervals that are biologically meaningful, we probed whether their methylation level correlates with the expression of nearby genes.

We first note that gene expression is more strongly correlated with the methylation of nearby VMRs than with the methylation of their promoters (Fig. 3a), indicating that VMR methylation is often a better predictor of gene expression than promoter methylation. Indeed, a gene-wise comparison revealed many genes whose expression is correlated with the nearest VMR but not with promoter methylation. One such example gene, *Htra1*, is depicted in Fig. 3b. While the promoter of this gene is lowly methylated regardless of gene expression, a VMR located downstream of the promoter is lowly methylated in cells with high *Htra1* expression.

**Improved identification of cell types.** We next tested whether our methods improve the ability to distinguish cell types and cell states (Fig. 4). To this end, we obtained cell type/state labels based on the single-cell

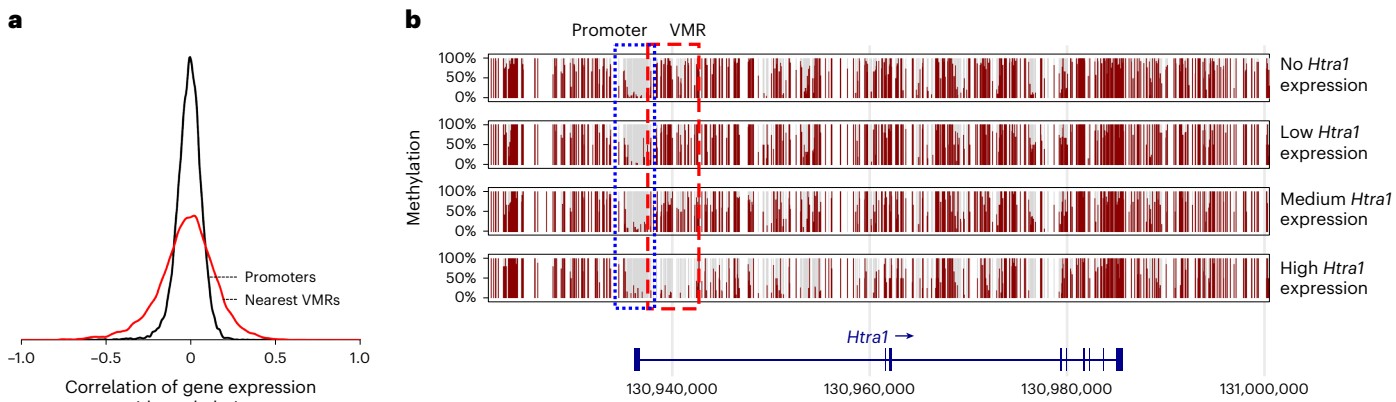

**Fig. 3 | Correlation of DNA methylation and gene expression. a**, The distribution of Pearson correlations between gene expression and promoter methylation (black) and gene expression and methylation of the nearest VMR (red). Promoters are defined as intervals ±2 kb around the TSS. **b**, The mean methylation near the gene *Htra1*. Cells are assigned to four groups based on the *Htra1* expression (group 0: cells with no *Htra1* expression; groups 1–3: cells that express *Htra1*, divided into three equally large groups with group 3 having the highest expression). Data from Kremer et al.[11]

transcriptomes from Kremer et al.[11] (Fig. 4a). We consider these cell labels as ground truth and tested whether we are able to distinguish the same groups of cells on the basis of their methylomes. To do this, we subjected the methylomes to various combinations of analysis methods including our own proposed methods and others that are commonly used. Specifically, we selected four different sets of genomic intervals at which CpG methylation is to be quantified: either VMRs detected with our approach, 100 kb genomic tiles, promoter regions (that is, transcription start site (TSS) ±2,000 bp) or candidate *cis*-regulatory elements (ENCODE cCREs[12]). To quantify methylation at these features, we either simply averaged in these intervals, obtaining methylation percentages, or calculated the shrunken mean of residuals, as described earlier. Finally, the resulting methylation matrices were subjected to iterative PCA (Methods) and subsequent UMAP for visualization.

Visual inspection of the resulting UMAPs revealed that our proposed combination of methods results in more clearly separated cell types, compared to a UMAP obtained with default analysis methods (Fig. 4b,c). While all cells form a continuous point cloud when using default methods, our improvements led to a clear separation of oligodendrocytes, oligodendrocyte precursor cells and endothelial cells. Furthermore, even cellular substates of cells in the continuous neural stem cell lineage were partially separated. To quantify this performance gain in a more rigorous manner, we used a score that quantifies whether cells were placed, in 15-dimensional principal component (PC) space, in a neighborhood comprising cells of the same cell type ('neighbor score'; Fig. 4e, see Methods for details). A higher neighbor score implies better separation of cell types inferred from single-cell transcriptomes of the same cells.

The neighborhood relation in PCA space is relevant as it is used in many downstream analyses, for example, for clustering. We ask how the neighbor score depends on the number of available cells. This is relevant as scBS protocols are costly and labor-intensive, and only few laboratories are currently able to obtain thousands of single-cell methylomes. To simulate smaller datasets, we subsampled the 1,566-cell dataset into smaller data sets, analyzed them again with all combinations of analysis methods, and calculated the mean neighbor score for each (Fig. 4d). The results confirmed that quantifying methylation at VMRs leads to a cleaner separation of cell states than using promoter regions or 100 kb tiles. Using the shrunken mean of residuals as a measure of DNA methylation improved the results further. This effect was most noticeable when quantifying promoters or genomic tiles, presumably because an individual promoter region or tile might span genomic regions with varying levels of DNA methylation as depicted in Fig. 1b, which our method accounts for. Although cell cluster separation

generally becomes more difficult in such cases, performance gains were observed also in smaller datasets.

Overall, VMR quantification yielded results similar to those obtained when quantifying ENCODE regulatory regions, even though the number of detected VMRs (63,421) is considerably smaller than the number of ENCODE cCREs (339,815). When we repeated our analysis using only the 63,421 cCREs with the highest coverage, the ability to distinguish cell types was diminished, suggesting that the average VMR is more informative for this task then the average cCRE (Extended Data Fig. 1a,b). This demonstrates the ability of our de novo VMR detection approach to identify in scBS data a parsimonious set of relevant elements. The overlap between VMRs and cCREs is limited (Fig. 4f), indicating that VMR detection yields information that is complementary to other epigenetic marks. A further benefit of VMR detection is that this approach is available even in the absence of such annotations, for instance when studying species other than human or mouse. Lastly, using VMRs over regulatory regions resulted in decreased RAM requirements as well as a shorter runtime, even when accounting for the additional step of VMR detection (Extended Data Fig. 1c,d).

We repeated this benchmark on an additional three published single-cell methylome datasets (Extended Data Fig. 2). These include neuronal subtypes of the murine cortex[8] (using cell type labels derived from CH methylation in genomic tiles instead of CpG methylation as ground truth), cells isolated from mouse embryos during the onset of gastrulation[10] (using RNA-derived cell clusters or alternatively embryonic stage as ground truth) and human colorectal cancer cells[13] (using sampling region as ground truth). Again, we subjected each dataset to all possible combinations of genomic feature selection and methylation quantification. We furthermore included three additional approaches to perform dimensionality reduction in our benchmarks, including PCA with two different preprocessing strategies, as well as Multi-Omics Factor Analysis version 2 (MOFA+), a dimensionality reduction technique designed for multimodal single-cell data that can also process methylation data[14].

These extensive benchmarks confirmed that our proposed combination of methods, that is, using the shrunken means of residuals of VMRs for dimensionality reduction, is able to distinguish diverse cellular properties such as cell type, colorectal cancer stage (normal tissue, primary tumor and metastasis), embryonic stage and germ layer.

**Robustness to parameter changes.** Next, we assessed whether our proposed workflow requires fine-tuning of parameters (Extended Data Fig. 3). To this end, we re-analyzed two datasets, as well as subsamples of the data with different VMR detection parameters, namely the width

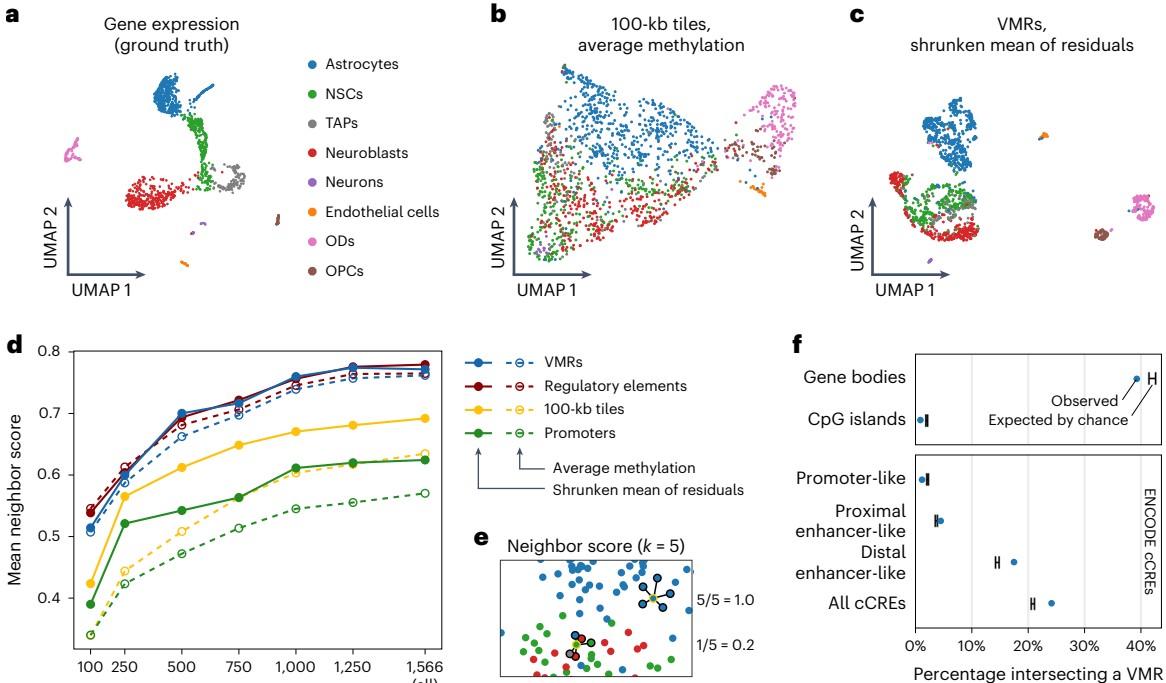

**Fig. 4 | A benchmark of our methods on single-cell multi-omic data of cells of the murine forebrain. a**, Cell labels based on clustering of single-cell transcriptomes from Kremer et al.[11]. OPCs, oligodendrocyte progenitor cells; TAPs, transit-amplifying progenitors. **b,c**, Exemplary UMAPs obtained when analyzing the dataset with conventional methods based on genome tiling (**b**) or our proposed methods (**c**). **d,e**, The mean degree of cell type separation (the neighbor score (**e**) computed in 15-dimensional PC space) obtained when analyzing single-cell methylomes with different combinations of methods (**d**).

Either VMRs, ENCODE regulatory elements, 100 kb genomic tiles or promoter regions (TSS ± 2 kb) were subjected to iterative PCA and UMAP. The CpG methylation in these intervals was quantified by either averaging (dashed lines) or using the shrunken mean of the residuals as proposed in this work (solid lines). The full 1,566-cell dataset was subsampled to simulate smaller datasets (*x* axis). **f**, The proportion of VMRs (blue dots) that have at least 1 bp overlap with other genomic features. The black range illustrates the minimum and maximum overlap observed for 100 reshufflings of the VMRs to randomly chosen positions.

of the sliding window in bp (set with the option '--bandwidth' in our MethSCAn software, default 2,000), the variance threshold above which windows are merged to VMRs ('--var-threshold', default 0.02) and the step size of the sliding window ('--stepsize', default 100 bp). This parameter sweep showed that our workflow gives good results over a wide range of parameter values. For the CpG methylation data of Luo et al.[8], the results are nearly independent of the parameters (Extended Data Fig. 3b). In the more challenging dataset of Kremer et al.[11], cell types were less cleanly separated when very large bandwidths, very strict variance thresholds or a very large step size was selected (Extended Data Fig. 3a,c). However, very small bandwidths or very lenient thresholds resulted in a much higher number of VMRs and thus long computing times. Overall, our default parameter combination provided good results and fast compute times in both datasets.

**Further applications.** Lastly, we asked whether our methods are also suitable for the analysis of DNA methylation outside the default CpG context. To this end, we revisited the Luo et al.[8] dataset but this time only considered CH methylation. VMR detection with default options produced results that were qualitatively similar to those reported in Luo et al.[8], suggesting that our methods are also suitable for this data type (Extended Data Fig. 4a). Finally, as single-cell methylome datasets are expected to rapidly grow in size in the coming years, we furthermore performed a stress test on a large dataset comprising 100,350 cells[15] (Extended Data Fig. 4b).

## Finding DMRs
A common task in the analysis of bulk bisulfite-sequencing data is the detection of DMRs between conditions, tissues or cell types[16,17]. As DNA methylation affects gene expression, DMRs can provide insights into

the unique epigenetic and gene regulatory characteristics of cell types. However, to date, no approach to detect DMRs in scBS data has been reported. To enable DMR detection in scBS data, we thus developed an approach that detects DMRs of variable size between two groups of cells and controls the false discovery rate (FDR) (Fig. 5a).

Similar to the previously described approach for VMR identification (Fig. 2), we divide each chromosome into overlapping windows shifted by a small and fixed size (step size) and quantify the methylation of each cell in each window. Next, instead of the variance, we obtain the *t* statistic as a measure of differential methylation between the two cell groups. We identify the windows with the most extreme *t* statistics, for example, windows in the 2% upper and lower tails. We then merge any overlapping windows in the upper tail into larger DMRs and do the same for windows in the lower tail, and then we recalculate the *t* statistic for each larger DMR.

To assess the statistical significance of DMRs, we repeat the same procedure on permutations of the scBS data, that is, the same dataset with randomly shuffled cell labels. The DMR *t* statistics obtained from permuted data are then used to estimate the FDR, yielding an adjusted *P* value for each DMR. While the primary purpose of VMRs is to provide better input for PCA and distance calculations, DMR detection facilitates the discovery of epigenetic differences between conditions or cell types, as we demonstrate next.

## Detecting DMRs between oligodendrocytes and neural stem cells
We used the single-cell multi-omics dataset[11] to evaluate our DMR detection approach. We first selected two cell populations from the healthy murine ventricular–subventricular zone: neural stem cells (NSCs, 130 cells) and oligodendrocytes (58 cells). Then, we used the

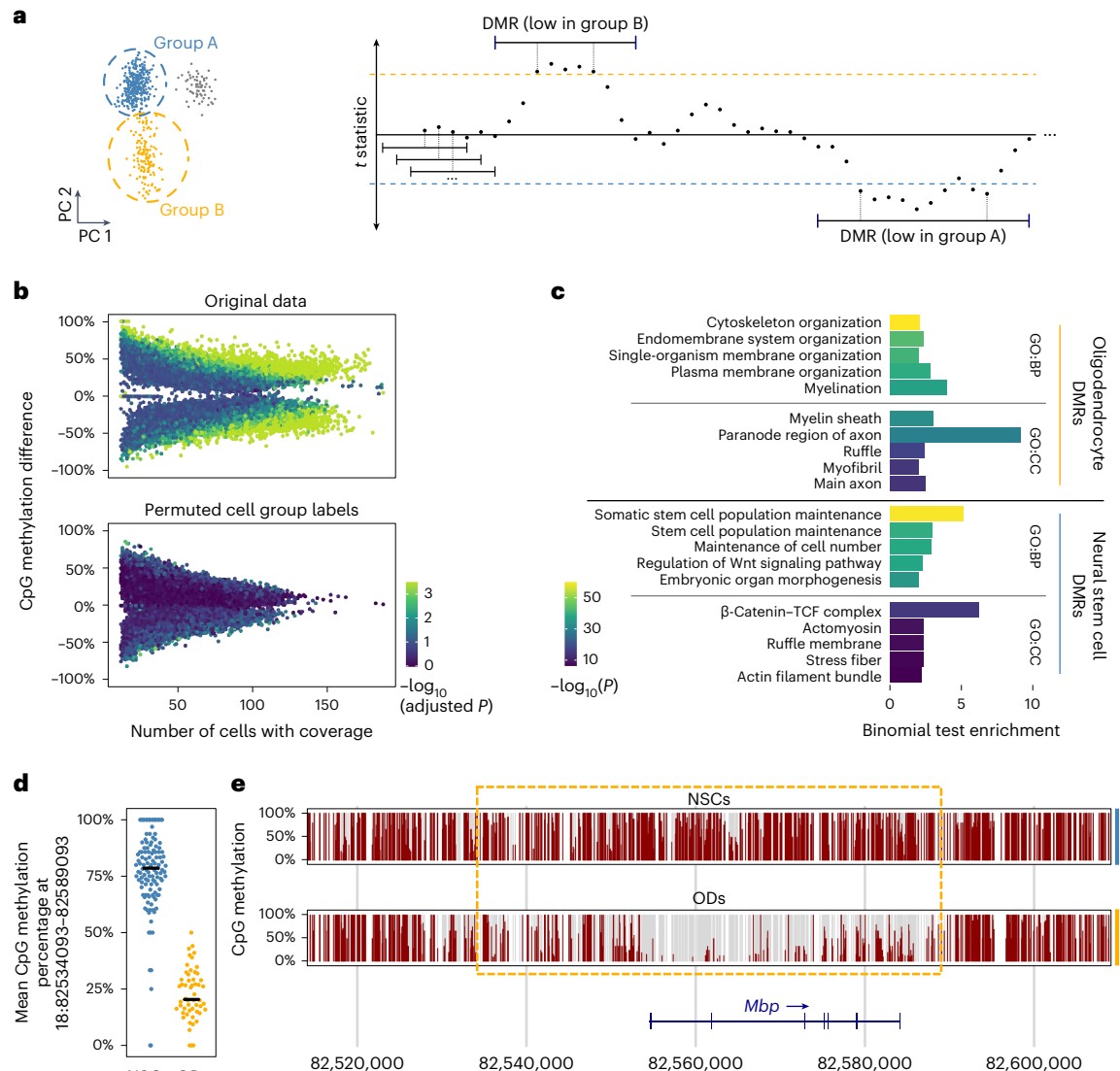

**Fig. 5 | Detection of DMRs. a**, A schematic depiction of our DMR detection algorithm. Points represent windows for which the *t* statistic is obtained, while dashed lines represent upper and lower *t* statistic thresholds. Windows exceeding either threshold are merged if they overlap, yielding DMRs that are lowly methylated in either group of cells. **b**, Top: DMRs detected between 58 oligodendrocytes (ODs) and 130 NSCs from Kremer et al.[11]. Bottom: DMRs detected in the same data with randomly permuted cell labels (bottom, used to estimate the FDR and determine adjusted *P* values). **c**, The enrichment of GO terms associated with DMRs lowly methylated in ODs (top) or NSCs (bottom). Depicted are the top five GO terms of the 'Biological Process' (GO:BP) and 'Cellular Component' (GO:CC) GO category, and their binomial test *P* value (two-sided, not adjusted for multiple comparisons) and enrichment, as reported by GREAT[18]. **d**, The mean methylation of NSCs and ODs at an exemplary DMR. Each point corresponds to a cell. Black lines denote the median. **e**, A detailed view of the DMR (yellow dashed rectangle) from **c** in pseudo-bulk samples consisting of NSCs or ODs. Vertical bars represent CpG sites.

method just described to identify DMRs between these two cell types (Fig. 5b). Repeating this after permuting the cell-type labels, in order to obtain a null distribution of the *t* statistics, yielded DMRs with much weaker methylation differences. Consequently, we could assign to many of the DMRs detected in the unpermuted data a low adjusted *P* value (Fig. 5b, colors).

Gene Ontology (GO) enrichment with the Genomic Regions Enrichment of Annotations Tool (GREAT)[18] revealed that DMRs lowly methylated in oligodendrocytes are located near genes involved in myelination, the main function of oligodendrocytes. Similarly, DMRs specifically demethylated in NSCs occur near genes involved in stem cell population maintenance. This demonstrates that our DMR detection approach is able to identify biologically meaningful DMRs, even in scBS datasets of modest cell number. Figure 5d,e depicts an exemplary DMR, located at the gene encoding myelin-basic protein (*Mbp*), the major component of myelin that is essential for myelination of neuronal

axons[19]. Our results suggest that oligodendrocyte-specific gene expression of *Mbp* is supported by low methylation at the detected DMR.

**The MethSCAn software toolkit**

We have implemented the methods just described in a Python package with a command line interface, MethSCAn (Single-Cell Analysis of Methylation data), which also offers a number of other functionalities for the analysis of scBS data. Figure 6 illustrates a typical scBS data analysis with MethSCAn and provides an overview of the implemented core functionalities. A tutorial that showcases the analysis of a small example dataset can be found at https://anders-biostat.github.io/MethSCAn.

The starting point of such an analysis are methylation files generated by tools such as Bismark[20], methylpy[21] or bisulfite-seq command line user interface toolkit[22]. Since it is inconvenient to work with hundreds or thousands of text files, MethSCAn provides the 'prepare' command that parses these methylation files and stores their content

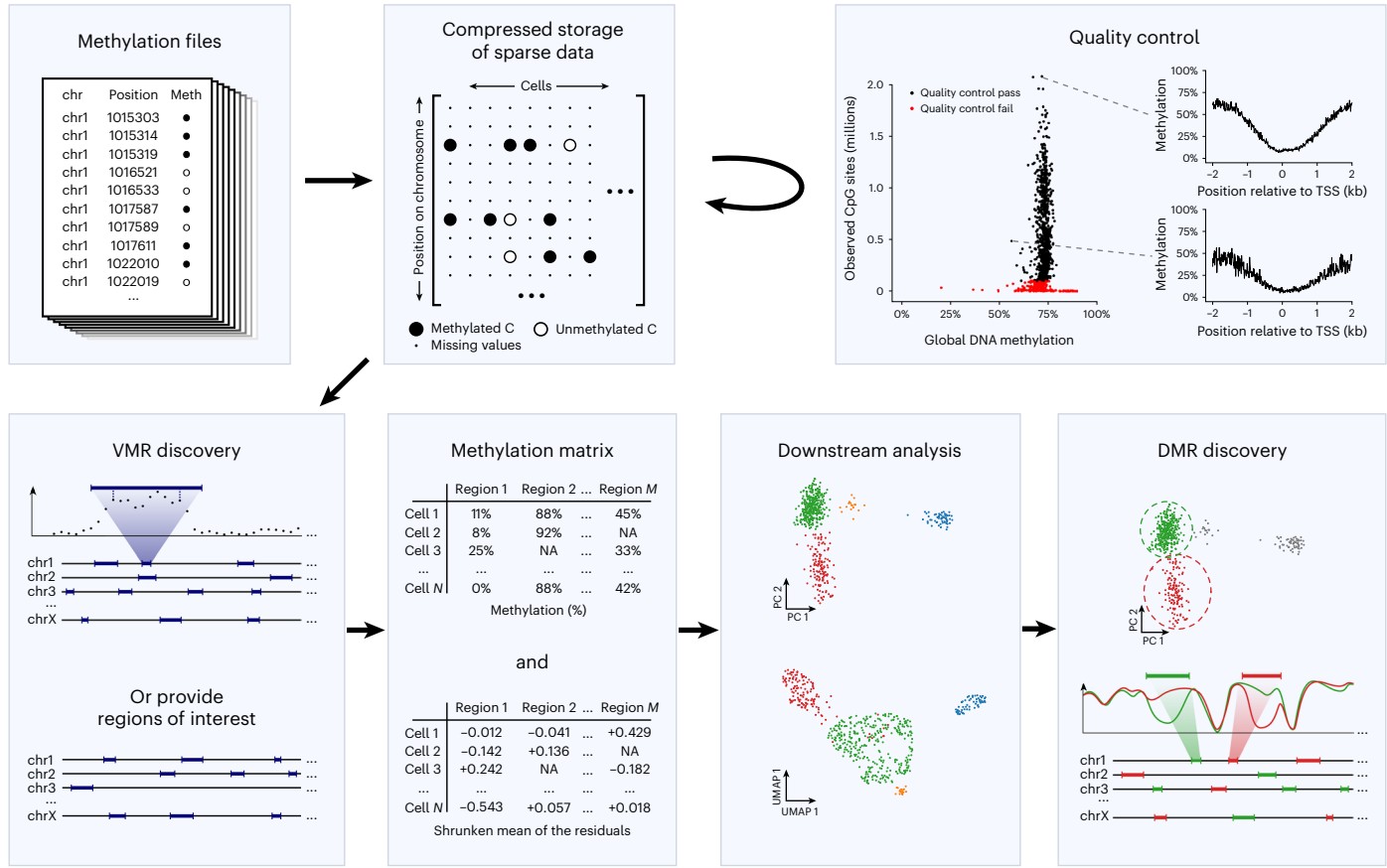

**Fig. 6 | An overview of the functionalities implemented in the MethSCAn package.** An scBS data analysis workflow that starts from methylation report files generated by mappers designed for bisulfite-converted sequencing reads. MethSCAn facilitates exploration of this data by constructing a methylation matrix suitable for dimensionality reduction and by enabling the detection of DMRs between groups of cells. Chr, chromosome; Meth, methylation status; NA: not available (missing values).

in a compressed format that enables efficient access to all CpG sites of the genome (Methods). The 'methscan prepare' command also computes a number of summary statistics for each cell, including the mean genome-wide methylation level and the number of observed CpG sites, that is, the number of CpG sites that have sequencing coverage. These summary statistics can be used to detect cells with poor quality. The quality of each single cell methylome can furthermore be inspected with 'methscan profile', which computes the average methylation profile of a set of user-defined genomic regions such as TSS at single-cell resolution. The TSS profile is a useful quality control plot since methylation shows a characteristic dip roughly ±1 kb around the TSS in mammalian genomes. Cells that do not show this pattern, or cells with few observed CpG sites, may then be discarded with 'methscan filter'.

After quality control, the user has access to the genome-wide VMR detection approach described earlier, using the 'methscan scan' command. This produces a browser extensible data (BED) file that lists the genome coordinates of VMRs. To finally obtain a methylation matrix analogous to a scRNA-seq count matrix, this BED file can be used as input for 'methscan matrix', which quantifies the methylation at genomic intervals in all cells. The command produces both the simple percentage of methylated CpG sites as well as our proposed methylation measure, that is, the shrunken mean of the residuals, which is more robust to variation in sequencing coverage or stochastic differences in read position between cells. We note that both 'matrix' and 'profile' accept any valid BED file as input, which means that the user can quantify and visualize methylation at any set of genomic features of interest, such as promoters, enhancers or transcription factor binding sites, obtaining one profile plot per cell. The methylation

matrix produced by the 'matrix' function can then serve as input for established methods used for the exploration of single-cell data, such as dimensionality reduction and cell clustering.

Finally, after annotation and exploration of the dataset, the user may specify two groups of cells for DMR detection with 'methscan diff'. This command produces a BED file listing the genome coordinates, adjusted *P* values, as well as several other metrics associated with DMRs. These DMRs may then be associated with nearby genes, or used for GO enrichment with tools such as GREAT[18] to enable functional interpretation.

## Discussion

Here, we have proposed an improved strategy to preprocess single-cell bisulfite sequencing data. On the basis of the observation that incomplete read coverage of a genomic interval can lead to inaccurate methylation estimates, we suggest a scoring scheme that is aware of a read's local context rather than just treating all reads in an interval alike.

Furthermore, we show a way to pinpoint minimal regions of high variability across cells, which we call VMRs. Unlike other tools for scBS data analysis[23–25], which rely on the user to manually specify which genomic intervals should be quantified, MethSCAn implements an approach to discover these intervals in the data itself. This not only reduces noise and allows to focus only on the features that are important for the given dataset but also provides useful input for genomics-style analyses. Depending on the research question at hand, individual VMRs may be related to nearby genomic features such as gene bodies or known regulatory elements, or subjected to gene ontology and motif enrichment.

To furthermore aid interpretation of scBS data, we developed and implemented an algorithm for genome-wide detection of DMRs in

single-cell methylomes. FDR estimation via permutation allows us to report the statistical significance of each DMR. In a similar manner as VMRs, pinpointing the regions that differ between groups of cells and hence have a putative regulatory role aids biological interpretability. For instance, applying our approach to NSCs and oligodendrocytes demonstrated that the obtained DMRs locate near meaningful loci associated with cell-type specific functions.

We also presented an open-source software tool called MethS-CAn that provides an easy-to-use implementation of the described algorithms. It can start directly from the output of methylation callers such as Bismark[20], bisulfite-seq command line user interface toolkit[22] and methylpy[21] and produce a cell × region matrix. We suggest iterative PCA as an approach to map the count matrix to a reduced-dimensional space overcoming the abundance of missing values. Alternatively, one may use for this purpose established tools based on matrix factorization such as MOFA+[14] (included in our benchmarks) or Linked Inference of Genomic Experimental Relationships[26]. Once a low-dimensional embedding is obtained, one can switch to well-established methods from scRNA-seq analysis, including visualization tools such as t-SNE and UMAP, Leiden/Louvain clustering, pseudotime trajectory analyses, and so on, for example, by using either overall toolkits such as Seurat[6] and ScanPy/EpiScanPy[7,25] or any of the many tools available for specific tasks. By offering these functions, MethSCAn bridges a gap in the chain of existing tools that has so far hindered practitioners in their data analysis. Our implementation can handle datasets of various sizes up to a 100,000 cells (Extended Data Fig. 4b).

By re-analyzing four published datasets, we show that these improvements to data preprocessing help to increase signal and decrease noise, resulting in a more informative intermediate-dimensional representation of the data. As examples of practical benefits, we demonstrate that our preprocessing allows for better distinction of cell subtypes, especially for challenging datasets comprising cellular substates and lineage transitions or for datasets with small cell number.

In conclusion, we have presented powerful improvements to scBS data preprocessing and a software tool that implements these.

## Online content

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

## Methods

### Raw data

Let us write $x_{ij}$ for the methylation status of CpG $i$ in cell $j$. The index $i$ runs over all CpG positions present in the genome, and the index $j$ over all cells in the assay. We write $x_{ij} = 0$ if position $i$ was found to be unmethylated in cell $j$ by bisulfite sequencing, $x_{ij} = 1$ if it was methylated and $x_{ij} = $ NA if position $i$ was not covered by reads from cell $j$ and the methylation status is therefore not available (NA).

These values can be readily obtained from single-cell bisulfite sequencing data using tools such as Bismark.

If multiple reads from the same cell cover a position, these will typically be PCR duplicates of each other and hence agree. Of course, the two alleles of a CpG may rarely differ in their methylation status. While it is, in principle, possible that one obtains discordant reads stemming from the same position on both the paternal and the maternal chromosomes of the same cell, this is so unlikely that we can ignore such cases. Hence, whenever the methylation caller reports multiple reads covering the same position in the same cell, we set $x_{ij}$ to 0 or 1 whenever all reads agree. When there is disagreement, we put $x_{ij} = $ NA by default, or optionally follow the majority of reads whenever possible.

For later use, we define $C$ as the set of all cells in the assay (that is, $C$ is the index set for the cell indices $j$). Moreover, we define $C_i \subset C$ as the set of all those cells $j$ that have reads covering position $i$:

$$C_i = \{j \in C : x_{ij} \neq \text{NA}\}.$$

Conversely, we define $G_j$ as the set of all the CpG positions $i$ covered by reads from cell $j$:

$$G_j = \{i : x_{ij} \neq \text{NA}\}.$$

### Data storage

The function 'methscan prepare' reads a set of methylation files (for example, produced by Bismark) and produces one file per chromosome. These files store the matrix $x$, where each column represents a cell and each row represents a base pair, in a space-efficient format as follows: $x$ is represented as a SciPy sparse matrix[27], encoding the actual values 0, 1 and NA as −1, 1 and 0, respectively. Since the vast majority of values in this matrix are missing owing to the sparsity of scBS data (and because rows for base pairs not corresponding to a CpG site contain no data), we encode missing values as zero and then store the data in compressed sparse row format. This format does not explicitly store zeroes (here, missing values) and is optimized for row-wise access, which results in substantial compression and allows fast access to the methylation status of genomic intervals. In all that follows here, any mention of $x$ will, however, always mean the encoding as $x_{ij} \in \{0, 1, \text{NA}\}$.

### Smoothing

For each CpG position $i$, we write

$$\bar{x}_i = \langle x_{ij} \rangle_{j \in C_i} = \frac{1}{|C_i|} \sum_{j \in C_i} x_{ij}$$

for the average methylation at position $i$, where $\langle \cdot \rangle$ denotes averaging, and the average runs over all the cells $j \in C_i$, that is, over those cells for which methylation data are available for position $i$.

We then run a kernel smoother over these per-position averages to obtain the smoothed averages $\tilde{x}_i$. Specifically, we use

$$\tilde{x}_i = \frac{\sum_{i'} \bar{x}_{i'} \, k_h(d_{ii'})}{\sum_{i'} k_h(d_{ii'})},$$

that is, $\tilde{x}_i$ is the weighted average over the per-position averages $\bar{x}_{i'}$, taken over the CpG sites $i'$ in the neighborhood of $i$, and weighted

using a smoothing kernel $k_h$ with bandwidth $h$. Here, $d_{ii'}$ is the distance between CpG positions $i$ and $i'$, measured in base pairs, $h$ is the smoothing bandwidth in base pairs (by default, $h = 1{,}000$) and $k_h$ is the tricube kernel

$$k_h(d) = \begin{cases} \left(1 - |d/h|^3\right)^3 & \text{for } |d| < h \\ 0 & \text{otherwise.} \end{cases}$$

### Methylation for an interval

Next, we discuss averaging methylation over a range of CpG sites.

Given an interval $I$ on the chromosome, we wish to quantify the average methylation $m_{Ij}$ of the CpG sites within the interval for cell $j$. If we interpret $I$ as the set of CpG positions $i$ in the interval, we may write

$$m_{Ij} = \langle x_{ij} \rangle_{i \in I \cap G_j}.$$

Here, the average runs over all those sites $i$ that lie within the interval $I$ and are covered by reads from cell $j$.

If we wish to compare cells, it can be helpful to center this quantity by subtracting its average using

$$z_{Ij} = m_{Ij} - \langle m_{Ij'} \rangle_{j' \in C}.$$

As an alternative, we suggest to consider the residuals of the individual CpG methylation values $x_{ij}$ from the smoothed average $\tilde{x}_i$

$$r_{ij} = x_{ij} - \tilde{x}_i,$$

and averaging over these, obtaining

$$r_{Ij} = \frac{1}{|I \cap C_i| + 1} \sum_{i \in I \cap C_i} (x_{ij} - \tilde{x}_i). \tag{1}$$

This is a shrunken average, with denominator $n + 1$. This extra pseudocount has the effect of shrinking the value toward the 'neutral' value 0, with the shrinkage becoming stronger if the data are 'weak' because the number $|I \cap C_i|$ of positions covered by reads from cell $j$ is low. In the extreme case of none of the reads from cell $j$ covering $I$, the sum becomes 0 and the denominator 1; that is, $r_{Ij} = 0$ in this case.

### Finding VMRs

For any interval $I$, we denote by $v_I$ the variance of its residual averages $r_{Ij}$:

$$v_I = \frac{1}{|C_I|} \sum_{j \in C_I} \left( r_{Ij} - \langle r_{Ij'} \rangle_{j' \in C_I} \right)^2, \tag{2}$$

where the average runs only over the set $C_I = \bigcup_{i \in I} C_i$ of those cells $j$ that have reads covering interval $I$.

To find VMRs, we define intervals $I_1, I_2, \ldots$, all of the same width, and with stepwise increasing starts, then calculate $v_1, v_2, \ldots$ for these intervals. We then mark the intervals with the 2% highest variances. We take the union of all these intervals, split the union into connected components and call each component a VMR. Putting that last step in other words: We take all the intervals with variance in the top 2 percentile, fuse intervals that overlap and call the regions thus obtained the VMRs.

### Finding DMRs

When searching for DMRs, we compare two groups of cells, whose index sets we denote by $C_A$ and $C_B$. For a given interval $I$, we calculate the mean each of the mean shrunken residuals $r_{Ij}$ (equation (1)) over the cells $j$ in each of the two groups:

$$\mu_I^g = \langle r_{Ij} \rangle_{j \in g}, \qquad g = A, B.$$

We also calculate a variance as in equation (2):

$$v_I^g = \frac{1}{|C_g|-1} \sum_{j \in g} (r_{Ij} - \mu_I^g)^2, \qquad g = \text{A, B.}$$

From this, we calculate Welch's $t$ statistic as usual:

$$t_I = \frac{\mu_I^A - \mu_I^B}{\sqrt{\frac{v_I^A}{|C_A|} + \frac{v_I^B}{|C_B|}}}.$$

To find candidate DMRs, we again define overlapping and stepwise-shifted intervals $I_1, I_2, \ldots$ as for the VMRs and calculate $t$ statistics $t_1, t_2, \ldots$ for these. As before, we take the top 2 percentile of these values, fuse intervals that overlap and call the regions thus obtained candidate DMRs. We repeat the procedure for the bottom 2 percentile to get the candidate DMRs for the other sign.

Next, we need to check these candidate DMRs for statistical significance. We first remind the readers here that, as this is a within-sample analysis, cells, not samples, are the statistical unit. Therefore, a call as significant implies that the same DMR is likely to be called again if we repeated the analysis with another set of cells taken from the same biological sample, not that it would generalize to further samples. This fact, although often overlooked, is common to all within-sample analyses in the single-cell field, for example, also to the differential expression tests performed in scRNA-seq analyses to find marker genes that differentiate clusters.

It may seem that we could use the standard procedure for the Welch $t$-test here, that is, use the Welch–Satterthwaite formula to get an approximate degree of freedom and then calculate the tail probability of the corresponding $t$ distribution. However, this is unlikely to hold for two reasons: First, the Welch–Satterthwaite degrees of freedom are only based on the number of cells per group and do not account for the fact that the read coverage might vary from cell to cell. Second, the fusing of the DMRs obtained in the scanning step to obtain fused candidate DMRs would invalidate subsequent $P$ value-based adjustment for multiple testing.

Therefore, we have instead implemented a permutation procedure, which works as follows: We randomly shuffle the assignment of the cells in $C_A \cup C_A$ to either of the two groups and then rerun the whole procedure, that is, the scanning step, the DMR fusing and the calculation of $t$ values from the (potentially fused) candidate DMRs. This needs to be done for a sufficiently large number of permutations. Running through the whole genome for each permutation would be too computationally expensive. Instead, we go through the genome only once, but reshuffle the group labels every 2 Mb.

All the $t$ values obtained from this permutation procedure are taken together to obtain an empirical null distribution. Then, we can use this null distribution to control the FDR by applying the Benjamini–Hochberg procedure in its $P$ value-free form: Let us write $T$ for the set of all $t$ values obtained from the 'real' assignment of cells to group labels and $T_0$ from the set of all $t$ values obtained from the shuffled assignments, that is, the empirical null distribution. To adjust a specific $t$ value $t_i \in T$, we calculate

$$p_i^{\text{adj}} = \frac{|\{t_j \in T_0 \,||t_j| > |t_i|\}|/|T_0|}{|\{t_j \in T \,||t_j| > |t_i|\}|/|T|}.$$

In words, we calculate which fraction of the null $t$ values is greater than $t$ by absolute value, and which fraction of the real $t$ values is. The ratio gives us the FDR we should expect if we used the $t$ value as threshold.

## Calculating cell-to-cell distances
Given a set $\mathcal{V} = \{I_1^v, I_2^v, \ldots\}$ of intervals corresponding to VMRs, we get a relative methylation fraction $r_{ij}$ for each VMR $I_i^v$ and each cell $j$ from

equation (1). The matrix thus obtained can then be centered and used as input for a PCA. If we calculate the top $R$ principal components, we thus obtain for each cell $j$ an $R$-dimensional principal component vector $\mathbf{x}_j^P$. For any two cells $j$ and $j'$, we use the Euclidean distance $\|\mathbf{x}_j^P - \mathbf{x}_{j'}^P\|$ as the measure of dissimilarity of the two cells. Thus, the matrix of PC scores can be used as input to dimension reduction methods such as t-SNE or UMAP, and to clustering methods like the Louvain or Leiden algorithm, which require such a matrix as input to the approximate nearest neighbor finding algorithm that is their first step.

## PCA with iterative imputation
Whenever a region is not covered by any read in a cell, the corresponding data entry in the input data matrix for PCA will be missing. The standard approach to calculate PCA, commonly done using the implicitly restarted Lanczos bidiagonalization algorithm[28], is not suited to deal with missing data. We circumvent this issue by simply using the PCA itself to impute the missing value in an approach that we call 'iterative PCA'.

Let us write $A$ for the matrix to which the PCA is to be applied, with the features (here, regions) represented by the matrix rows. The matrix has already been centered, that is, $\sum_i a_{ij} = 0$. To establish notation, we remind the reader that performing a PCA on $A$ means finding the singular value decomposition $A = UDR^\top$, with $D$ diagonal and $U$ and $R$ orthonormal. The PCA scores are contained in $X = UD$, the loadings in $R$. To reconstruct the input data $A$ from the PCA representation, one may use $A = XR^\top$, that is, $a_{ij} = \sum_r x_{ir} r_{jr}$, where the equation is exact if $r$ runs over all principal components and approximate if it is truncated to the leading ones.

Our iterative imputation strategy is now simply the following: We first replace all missing values in the row-centered input matrix $A$ with zeroes and perform the (truncated) PCA. Then, we use the PCA predictions for the missing values, that is, the $a_{ij} = \sum_r x_{ir} r_{jr}$, as refined stand-ins for the missing values in $A$ and run PCA once more. This can now be iterated until convergence.

We note that similar approaches have also been used elsewhere[29].

## Analysis of scBS datasets for benchmarks
To analyze scBS data from Kremer et al.[11], single-cell CpG methylation reports from all conditions were first stored with 'methscan prepare' and then smoothed with 'methscan smooth' using the default bandwidth of 1,000 bp. These data were then analyzed multiple times with different combinations of analysis methods, namely four ways to divide the genome into intervals, two ways to quantify methylation in these intervals and four approaches for dimensionality reduction. The following four sets of genomic intervals were used: VMRs, obtained with 'methscan scan' using the current default options (bandwidth of 2,000, step size of 10, variance threshold of 0.02 and minimum cell requirement of 6); adjacent tiles of 100 kb width; promoter regions as defined by the ±2 kb domain around the TSS of coding genes; and candidate *cis*-regulatory regions annotated by the ENCODE consortium[12]. Methylation was quantified either by averaging binary methylation values, or by calculating the shrunken mean of the residuals as described earlier.

We used four different approaches for dimensionality reduction. Three of them involve imputation of missing values followed by PCA: The first approach, iterative PCA, was described earlier. Second, 'PCA on high-quality features' imputes missing values with the mean methylation level of a given interval, while only retaining high-quality features selected as in Luo et al.[8]: We selected tiles spanning ≥20 CpG sites and with sequencing coverage in at least 70% of cells. We then imputed missing values with the mean of each tile, centered the values and performed PCA. The third approach, 'mean-imputed PCA' is identical to the second approach but without the quality-filtering step. Lastly, we used MOFA+ with default parameters instead of PCA, which does not require imputation of missing values. In all four cases, we reduced the dimensionality of the input data to 15 PCs or MOFA factors. In some

cases, MOFA+ returned a smaller number of factors, since some of the requested 15 had zero variance. For visualization, these 15 PCs or factors were subjected to UMAP with parameters min_dist = 0.2 and init = 'spca'. To flexibly adapt to datasets of different sizes, we set n_neighbors = $\frac{\sqrt{n}}{1.5}$ (rounded to the nearest integer), where $n$ is the total cell number.

The same analysis was repeated for three additional scBS datasets[8,10,13] and for smaller datasets generated by randomly subsampling cells separately from these datasets.

VMRs that intersect protein-coding gene bodies, CpG islands (from the University of California Santa Cruz genome browser) or cCREs were quantified by subtracting VMRs with at least 1 bp of overlap using 'bedtools subtract -A'[30] and counting the remaining VMRs.

To test our DMR detection approach, we selected oligodendrocytes and NSCs from healthy wild-type mice of Kremer et al.[11] and ran 'methscan diff' with default parameters. For GO enrichment analysis, DMRs with adjusted $P < 0.01$ were uploaded to GREAT 4.0.4 (ref. [18]) with the association rule 'basal plus extension, 0 bp upstream, 20 kb downstream, 1 Mbp max extension, curated regulatory domains included'.

### Mean neighbor score

To assess the performance of our methods, we employed a score that quantifies how well cell types (or cell states) are separated in 15-dimensional PCA space. For data from Luo et al.[8], we used cell type labels that the authors manually curated based on CH methylation. For the multi-omic dataset, we repeated the single-cell transcriptomics analysis described in Kremer et al.[11] with two adjustments: We did not filter off-target cells such as endothelial cells, and we annotated cell types using Leiden clustering with the Seurat[6] function 'FindClusters(resolution = 0.1)'. The score, based on the $\Gamma$ score[31], varies between 0 and 1, where higher scores reflect better separation of cell types. For each cell $j$, we count how many of its $k$ nearest neighbors have been assigned to the same cell type as cell $j$. We denote this count by $a_j^k$, where we have chosen $k = \frac{\sqrt{n}}{1.5}$, rounded. The overall score used to evaluate a given combination of methods is then simply the mean of all cell-wise scores.

### Correlation of DNA methylation and gene expression

To assess the correlation between gene expression and the methylation status of VMRs or promoters, we first detected VMRs with 'methscan scan --bandwidth 1000 --var-threshold 0.05'. We then quantified DNA methylation at VMRs and promoters with 'methscan matrix'. We defined promoters as ±2,000 bp regions centered on a gene's TSS. When multiple TSSs were annotated, we chose the TSS of the 'principal' isoform[32]. Log-normalized expression values reported by Kremer et al.[11] were then correlated with methylation of the gene's promoter or with methylation of the VMR closest to the gene body. When multiple VMRs intersected the gene body, we chose the VMR with the highest methylation variance. As a measure of methylation, we used the shrunken mean of the residuals. We omitted lowly expressed genes (with scRNA-seq counts in <5% of cells) and promoters and VMRs with low scBS coverage (in <10 cells).

MethSCAn was implemented in Python 3.8 using the packages NumPy 1.20.1, SciPy 1.6.1, numba 0.53.0 and Pandas 1.2.3. Benchmarks were performed on MethSCAn version 0.6.2 using Snakemake 7.26 and analyzed/visualized with tidyverse 1.3.1 packages.

### Reporting summary

Further information on research design is available in the Nature Portfolio Reporting Summary linked to this article.

### Data availability

All data used to benchmark and showcase the MethSCAn software is publicly available under the following National Center for Biotechnology Information Gene Expression Omnibus accessions: single-cell multi-omics of the murine forebrain[11]: GSE210806; mouse neurons[8]: GSE97179; colorectal cancer[13]: GSE97693; mouse gastrulation[10]: GSE121708; 100k brain cells[15]: GSE132489.

### Code availability

MethSCAn is available on the Python Package Index and can be installed by using the Python package installer 'pip'. At https://anders-biostat.github.io/MethSCAn we provide detailed documentation including a link to the source code and a tutorial that demonstrates a complete MethSCAn analysis on an example dataset.

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

### Acknowledgements

S.A. acknowledges funding by the Klaus Tschira Foundation (project 00.022.2019). A.M.-V. acknowledges funding from the European Commission via ERC grant 771376, from the DFG via SFB 873 and from the DKFZ. We thank A. Uvarovskii for code refactoring and helpful suggestions on the source code.

### Author contributions

L.P.M.K., S.A. and A.M.-V. conceived the project, and L.P.K.M. and S.A. worked out the method. L.P.K.M. wrote the software package, with contributions from M.M.B. (differential methylation functionality) and L.K. L.P.M.K., S.O., L.K. and M.M.B. prepared the use-case demonstrations and performed the benchmarks. S.C. and A.M.-V. contributed experimental data and biological expertise. L.P.M.K., A.M.-V. and S.A. contributed supervision and project management. L.P.M.K. and S.A. wrote the paper.

### Funding

### Competing interests

The authors declare no competing interests.

### Additional information

**Extended data** is available for this paper at https://doi.org/10.1038/s41592-024-02347-x.

**Correspondence and requests for materials** should be addressed to Lukas P. M. Kremer, Ana Martin-Villalba or Simon Anders.

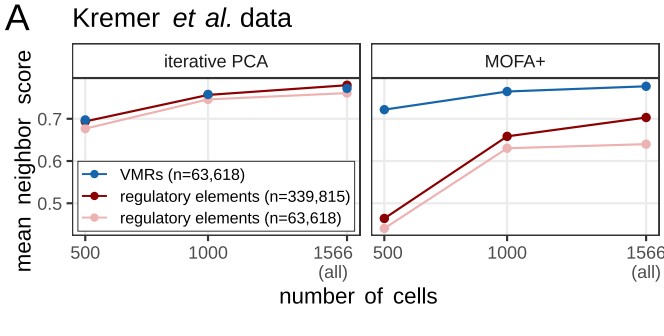

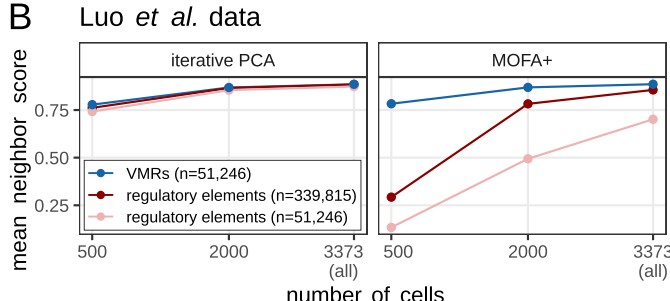

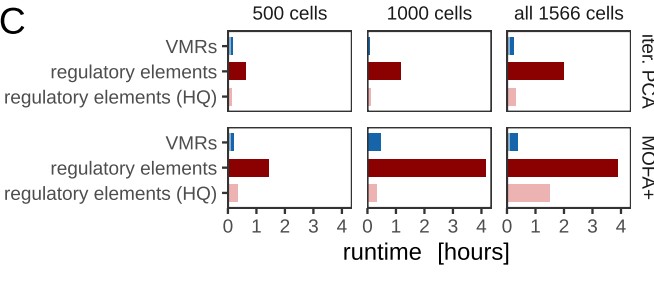

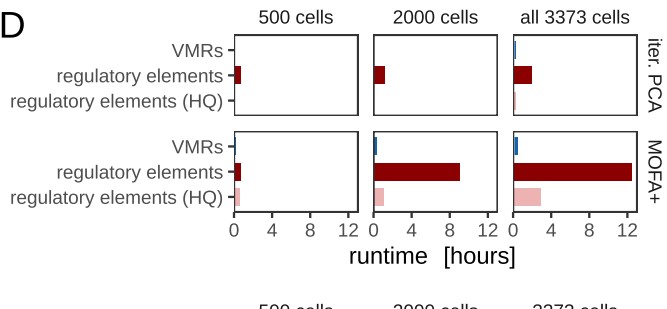

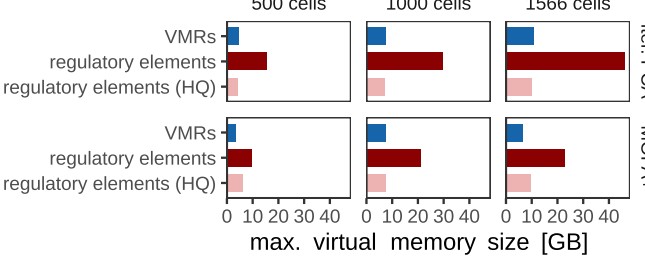

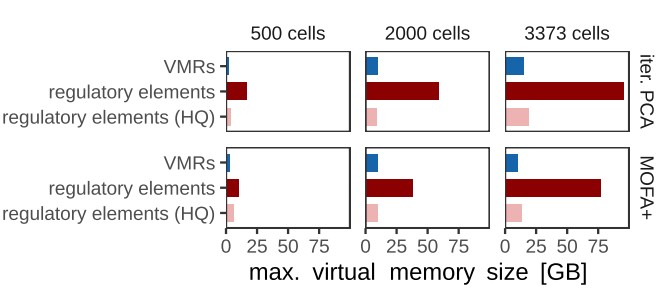

**Extended Data Fig. 1 | VMRs are more informative than an equal number of regulatory regions with high sequencing coverage. (A-B)** Mean neighbor score obtained when quantifying CpG methylation at VMRs, all ENCODE regulatory regions, or a subset of regulatory regions for the data sets of Kremer et al.[11] and Luo et al.[8]. The subset of high-quality (HQ) regulatory regions was selected in such a way that it matches the number of VMRs and contains those regulatory regions with the highest coverage. **(C-D)** Runtime (top) and RAM usage (bottom) of the two dimensionality reduction techniques shown in **A** and **B**.

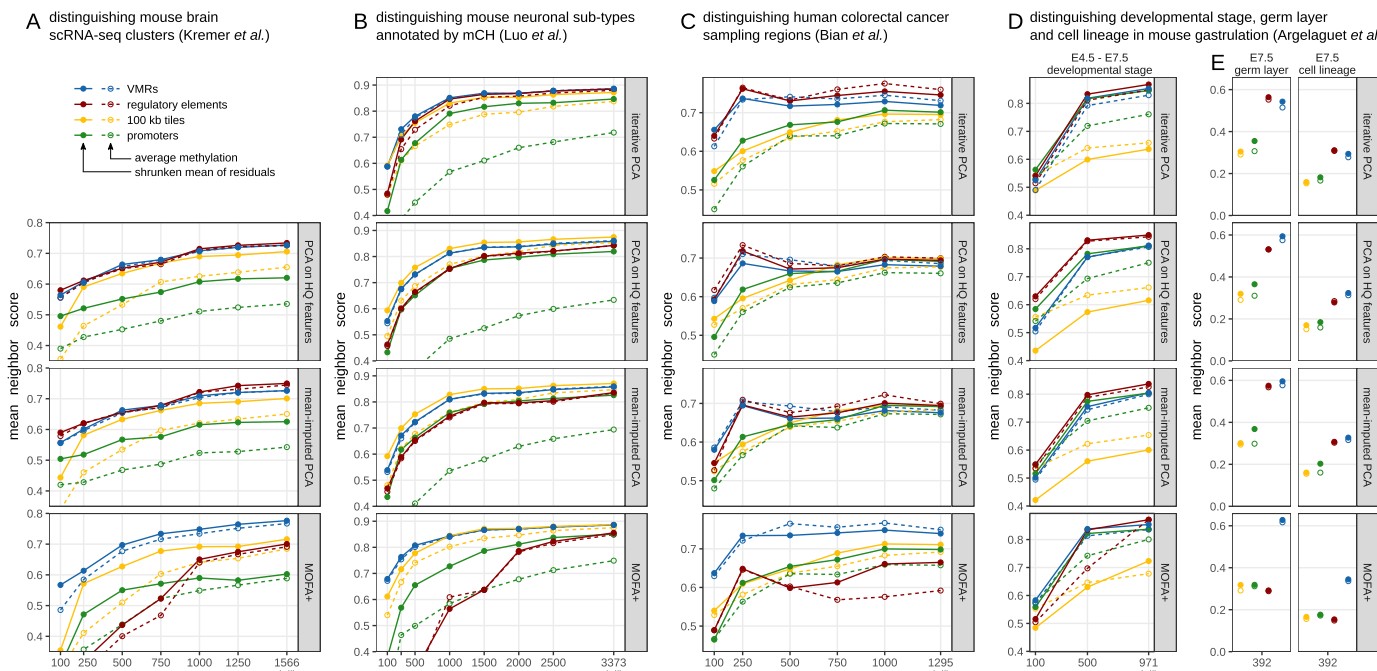

**Extended Data Fig. 2 | Benchmark of several combinations of analysis methods on four single-cell methylation data sets. (A-E)** Benchmarking the ability to separate groups of cells based on CpG methylation data, using different analysis approaches. Methylation matrices were obtained either by averaging CpG methylation in genomic intervals (dashed lines, hollow points) or using the shrunken mean of residuals (solid lines and points). Each analysis was performed for the full data sets and sub-samples of each. Four sets of genomic intervals (VMRs, ENCODE regulatory elements, 100 kb tiles or promoters) were quantified and separately subjected to four dimensionality reduction techniques (see Methods for details): iterative PCA as proposed by us, mean-imputed PCA on intervals with high sequencing coverage[8], mean-imputed PCA on all intervals, and MOFA+[14] on all intervals. For each combination of methods, we quantified the ability to use distance in 15-dimensional PCA space to separate ground-truth cell labels reported by the authors (neighbor score). **(A)** Separation of neural cell types/states annotated based on scRNA-seq of the same cells[11]. **(B)** Separation of neuron sub-types, annotated by averaging CH-methylation (mCH) in 100 kb genomic tiles[8]. **(C)** Separation of human colorectal cancer sampling regions (primary tumor, normal adjacent tissue, lymph node metastasis, liver metastasis before and after treatment, omental metastasis)[13]. **(D-E)** Separation of three properties of mouse embryo cells: The developmental stage (E4.5, E5.5, E6.5, E7.5), the germ layer or extra-embryonic tissue of origin, or the cell lineage. Germ layer and lineage (**E**, only E7.5-cells) were annotated by Argelaguet et al.[10] based on scRNA-seq of the same cells.

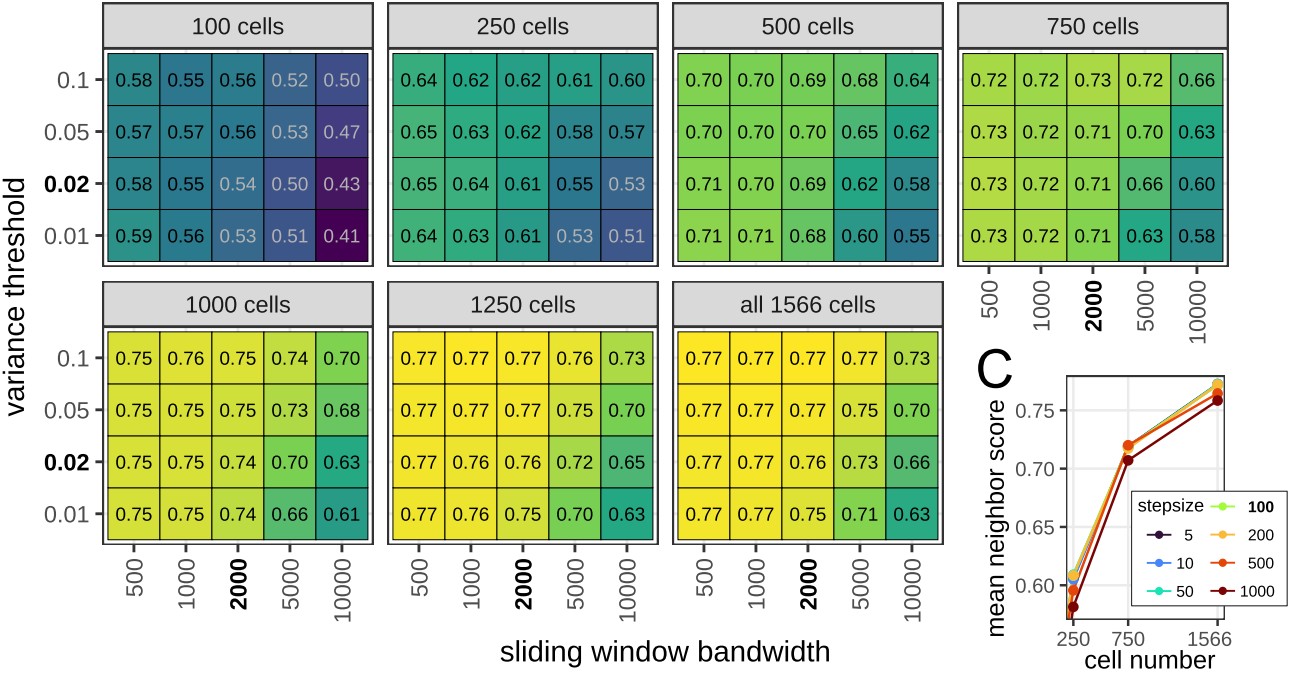

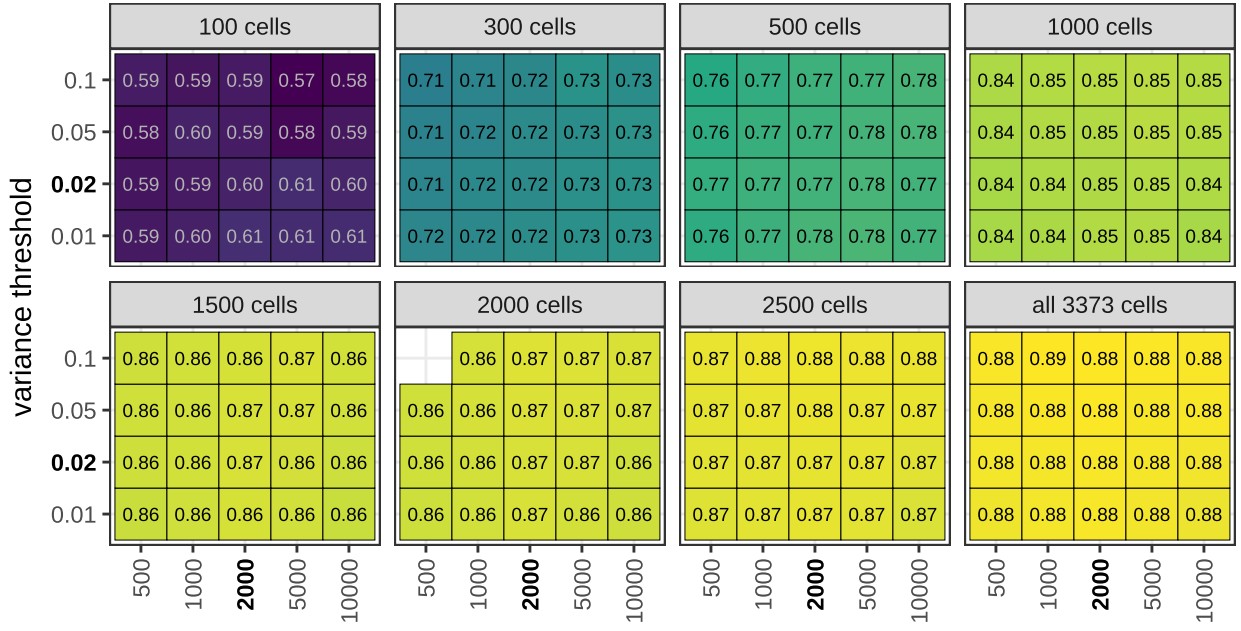

**Extended Data Fig. 3 | Effect of VMR detection parameters on separation of cell types. (A-B)** Mean neighbor scores obtained after analyzing single-cell methylomes from Kremer et al.[11] (**A**) or Luo et al.[8] (**B**) with our proposed methods. VMRs were detected with 'methscan scan' using various sliding window bandwidths, variance thresholds, and on sub-samples of the full data sets. (**C**) Mean neighbor scores obtained after analyzing the Kremer et al.[11] data set with various sliding window step sizes.

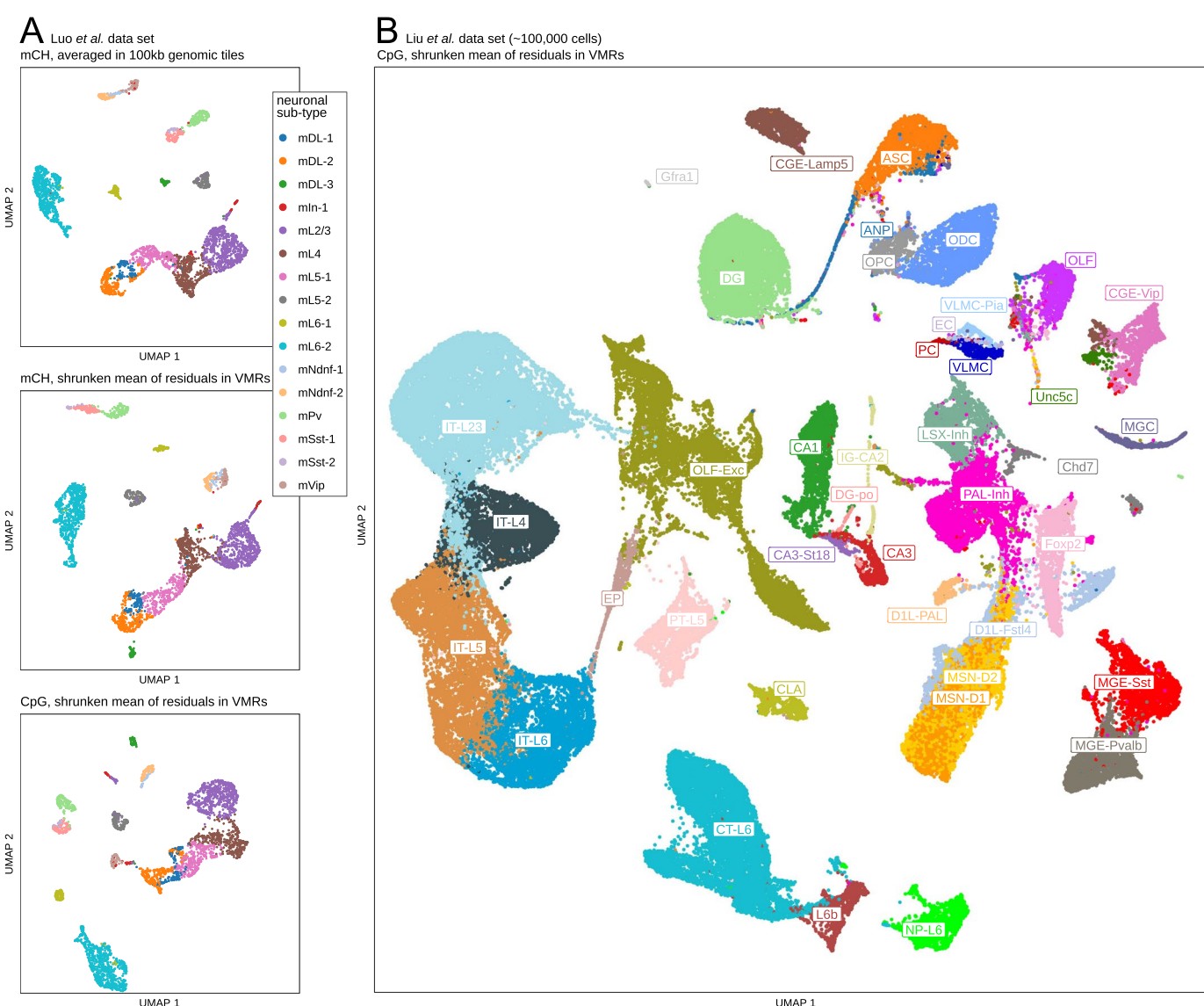

**A** Luo *et al.* data set
mCH, averaged in 100kb genomic tiles

neuronal sub-type
- mDL-1
- mDL-2
- mDL-3
- mIn-1
- mL2/3
- mL4
- mL5-1
- mL5-2
- mL6-1
- mL6-2
- mNdnf-1
- mNdnf-2
- mPv
- mSst-1
- mSst-2
- mVip

mCH, shrunken mean of residuals in VMRs

CpG, shrunken mean of residuals in VMRs

**B** Liu *et al.* data set (~100,000 cells)
CpG, shrunken mean of residuals in VMRs

**Extended Data Fig. 4 | MethSCAn performs well on CH-methylation data and on extremely large data sets. (A)** UMAPs obtained when analysing data from Luo et al.[8] with three different strategies for producing a methylation matrix: top: averaging CH-methylation (mCH) in 100 kb genomic tiles (as done in ref. [8] to obtain the depicted cell type labels); center: using the shrunken mean of residuals to quantify mCH in mCH-VMRs; bottom: shrunken mean of residuals to quantify CpG methylation in CpG-VMRs. **(B)** UMAP obtained when analysing 100 350 neural single-cell methylomes[15] with the default MethSCAn workflow. The depicted cell labels were reported by Liu et al.[15]. Analysing this large data set takes approximately a week on a computer with 256 GB RAM and 48 CPUs, with almost all of this time (152 hours) spent by the 'prepare' step, which is run only once per data set. The speed of this step can be increased considerably by processing chromosomes in parallel, which reduced the total runtime to approximately two days in our case.

Ana Martin-Villalba,
Lukas P.M. Kremer

# Reporting Summary

## Statistics

For all statistical analyses, confirm that the following items are present in the figure legend, table legend, main text, or Methods section.

| n/a | Confirmed | |
|---|---|---|
| ☒ | ☐ | The exact sample size (*n*) for each experimental group/condition, given as a discrete number and unit of measurement |
| ☐ | ☒ | A statement on whether measurements were taken from distinct samples or whether the same sample was measured repeatedly |
| ☐ | ☒ | The statistical test(s) used AND whether they are one- or two-sided<br>*Only common tests should be described solely by name; describe more complex techniques in the Methods section.* |
| ☐ | ☒ | A description of all covariates tested |
| ☐ | ☒ | A description of any assumptions or corrections, such as tests of normality and adjustment for multiple comparisons |
| ☒ | ☐ | A full description of the statistical parameters including central tendency (e.g. means) or other basic estimates (e.g. regression coefficient) AND variation (e.g. standard deviation) or associated estimates of uncertainty (e.g. confidence intervals) |
| ☐ | ☒ | For null hypothesis testing, the test statistic (e.g. *F*, *t*, *r*) with confidence intervals, effect sizes, degrees of freedom and *P* value noted<br>*Give P values as exact values whenever suitable.* |
| ☒ | ☐ | For Bayesian analysis, information on the choice of priors and Markov chain Monte Carlo settings |
| ☒ | ☐ | For hierarchical and complex designs, identification of the appropriate level for tests and full reporting of outcomes |
| ☒ | ☐ | Estimates of effect sizes (e.g. Cohen's *d*, Pearson's *r*), indicating how they were calculated |

*Our web collection on statistics for biologists contains articles on many of the points above.*

## Software and code

Policy information about availability of computer code

| Data collection | No software was used for data collection. |
|---|---|
| Data analysis | We developed a custom software, MethSCAn, to analyze single-cell bisulfite sequencing data. MethSCAn is free and open source. The package is available on the Python Package Index (PyPI) and can be installed with the Python package installer pip.The source code is hosted at https://github.com/anders-biostat/MethSCAn , where we also provide detailed documentation including a tutorial that demonstrates a complete MethSCAn analysis on an example data set. Benchmarks were performed using MethSCAn version 0.6.2. Note that the MethSCAn package was called 'scbs' before publication. The source code of legacy MethSCAn / scbs versions < 1.0 are hosted at https://github.com/LKremer/scbs .<br>Other software that was used:<br>- Python 3.8<br>- R 4.1.0<br>- numba 0.53.0<br>- SciPy 1.6.1<br>- NumPy 1.20.1<br>- Pandas 1.2.3<br>- tidyverse 1.3.1<br>- GREAT 4.0.4<br>- Snakemake 7.26<br>- IRLBA 2.3.5.1 |

For manuscripts utilizing custom algorithms or software that are central to the research but not yet described in published literature, software must be made available to editors and reviewers. We strongly encourage code deposition in a community repository (e.g. GitHub). See the Nature Portfolio guidelines for submitting code & software for further information.

## Data

Policy information about availability of data

All manuscripts must include a data availability statement. This statement should provide the following information, where applicable:

- Accession codes, unique identifiers, or web links for publicly available datasets
- A description of any restrictions on data availability
- For clinical datasets or third party data, please ensure that the statement adheres to our policy

All data is used to benchmark and showcase the MethSCAn software is publicly available. The single-cell multi-omic data of cells of the murine forebrain (Kremer, 2022) is available at the NCBI Gene Expression Omnibus (GEO) under the accession GSE210806. A data availibility statement with this accession has been added to the manuscript.

## Research involving human participants, their data, or biological material

Policy information about studies with human participants or human data. See also policy information about sex, gender (identity/presentation), and sexual orientation and race, ethnicity and racism.

| Reporting on sex and gender | n/a |
|---|---|
| Reporting on race, ethnicity, or other socially relevant groupings | n/a |
| Population characteristics | n/a |
| Recruitment | n/a |
| Ethics oversight | n/a |

Note that full information on the approval of the study protocol must also be provided in the manuscript.

# Field-specific reporting

Please select the one below that is the best fit for your research. If you are not sure, read the appropriate sections before making your selection.

☒ Life sciences　　　☐ Behavioural & social sciences　　　☐ Ecological, evolutionary & environmental sciences

For a reference copy of the document with all sections, see nature.com/documents/nr-reporting-summary-flat.pdf

# Life sciences study design

All studies must disclose on these points even when the disclosure is negative.

| Sample size | Since we did not perform experiments, this point does not apply. We benchmarked our software on four distinct data sets from different biological contexts (embryo, brain, stem cells, cancer) to show that our software performs well on diverse data sets. |
|---|---|
| Data exclusions | As detailed in the methods, we filtered low-quality cells and low-quality genomic intervals before dimensionality reduction. |
| Replication | We did not perform experiments, hence no replication was used. |
| Randomization | We did not perform experiments, hence no randomization was used. |
| Blinding | We did not perform experiments, hence no blinding was used. |

# Behavioural & social sciences study design

All studies must disclose on these points even when the disclosure is negative.

| Study description | Briefly describe the study type including whether data are quantitative, qualitative, or mixed-methods (e.g. qualitative cross-sectional, quantitative experimental, mixed-methods case study). |
|---|---|
| Research sample | State the research sample (e.g. Harvard university undergraduates, villagers in rural India) and provide relevant demographic information (e.g. age, sex) and indicate whether the sample is representative. Provide a rationale for the study sample chosen. For studies involving existing datasets, please describe the dataset and source. |

| | |
|---|---|
| Sampling strategy | *Describe the sampling procedure (e.g. random, snowball, stratified, convenience). Describe the statistical methods that were used to predetermine sample size OR if no sample-size calculation was performed, describe how sample sizes were chosen and provide a rationale for why these sample sizes are sufficient. For qualitative data, please indicate whether data saturation was considered, and what criteria were used to decide that no further sampling was needed.* |
| Data collection | *Provide details about the data collection procedure, including the instruments or devices used to record the data (e.g. pen and paper, computer, eye tracker, video or audio equipment) whether anyone was present besides the participant(s) and the researcher, and whether the researcher was blind to experimental condition and/or the study hypothesis during data collection.* |
| Timing | *Indicate the start and stop dates of data collection. If there is a gap between collection periods, state the dates for each sample cohort.* |
| Data exclusions | *If no data were excluded from the analyses, state so OR if data were excluded, provide the exact number of exclusions and the rationale behind them, indicating whether exclusion criteria were pre-established.* |
| Non-participation | *State how many participants dropped out/declined participation and the reason(s) given OR provide response rate OR state that no participants dropped out/declined participation.* |
| Randomization | *If participants were not allocated into experimental groups, state so OR describe how participants were allocated to groups, and if allocation was not random, describe how covariates were controlled.* |

# Ecological, evolutionary & environmental sciences study design

All studies must disclose on these points even when the disclosure is negative.

| | |
|---|---|
| Study description | *Briefly describe the study. For quantitative data include treatment factors and interactions, design structure (e.g. factorial, nested, hierarchical), nature and number of experimental units and replicates.* |
| Research sample | *Describe the research sample (e.g. a group of tagged Passer domesticus, all Stenocereus thurberi within Organ Pipe Cactus National Monument), and provide a rationale for the sample choice. When relevant, describe the organism taxa, source, sex, age range and any manipulations. State what population the sample is meant to represent when applicable. For studies involving existing datasets, describe the data and its source.* |
| Sampling strategy | *Note the sampling procedure. Describe the statistical methods that were used to predetermine sample size OR if no sample-size calculation was performed, describe how sample sizes were chosen and provide a rationale for why these sample sizes are sufficient.* |
| Data collection | *Describe the data collection procedure, including who recorded the data and how.* |
| Timing and spatial scale | *Indicate the start and stop dates of data collection, noting the frequency and periodicity of sampling and providing a rationale for these choices. If there is a gap between collection periods, state the dates for each sample cohort. Specify the spatial scale from which the data are taken* |
| Data exclusions | *If no data were excluded from the analyses, state so OR if data were excluded, describe the exclusions and the rationale behind them, indicating whether exclusion criteria were pre-established.* |
| Reproducibility | *Describe the measures taken to verify the reproducibility of experimental findings. For each experiment, note whether any attempts to repeat the experiment failed OR state that all attempts to repeat the experiment were successful.* |
| Randomization | *Describe how samples/organisms/participants were allocated into groups. If allocation was not random, describe how covariates were controlled. If this is not relevant to your study, explain why.* |
| Blinding | *Describe the extent of blinding used during data acquisition and analysis. If blinding was not possible, describe why OR explain why blinding was not relevant to your study.* |

Did the study involve field work?  ☐ Yes   ☐ No

## Field work, collection and transport

| | |
|---|---|
| Field conditions | *Describe the study conditions for field work, providing relevant parameters (e.g. temperature, rainfall).* |
| Location | *State the location of the sampling or experiment, providing relevant parameters (e.g. latitude and longitude, elevation, water depth).* |
| Access & import/export | *Describe the efforts you have made to access habitats and to collect and import/export your samples in a responsible manner and in compliance with local, national and international laws, noting any permits that were obtained (give the name of the issuing authority, the date of issue, and any identifying information).* |
| Disturbance | *Describe any disturbance caused by the study and how it was minimized.* |

# Reporting for specific materials, systems and methods

We require information from authors about some types of materials, experimental systems and methods used in many studies. Here, indicate whether each material, system or method listed is relevant to your study. If you are not sure if a list item applies to your research, read the appropriate section before selecting a response.

## Materials & experimental systems

| n/a | Involved in the study |
|---|---|
| ☒ | ☐ Antibodies |
| ☒ | ☐ Eukaryotic cell lines |
| ☒ | ☐ Palaeontology and archaeology |
| ☒ | ☐ Animals and other organisms |
| ☒ | ☐ Clinical data |
| ☒ | ☐ Dual use research of concern |
| ☒ | ☐ Plants |

## Methods

| n/a | Involved in the study |
|---|---|
| ☒ | ☐ ChIP-seq |
| ☒ | ☐ Flow cytometry |
| ☒ | ☐ MRI-based neuroimaging |

## Antibodies

| | |
|---|---|
| Antibodies used | *Describe all antibodies used in the study; as applicable, provide supplier name, catalog number, clone name, and lot number.* |
| Validation | *Describe the validation of each primary antibody for the species and application, noting any validation statements on the manufacturer's website, relevant citations, antibody profiles in online databases, or data provided in the manuscript.* |

## Eukaryotic cell lines

Policy information about cell lines and Sex and Gender in Research

| | |
|---|---|
| Cell line source(s) | *State the source of each cell line used and the sex of all primary cell lines and cells derived from human participants or vertebrate models.* |
| Authentication | *Describe the authentication procedures for each cell line used OR declare that none of the cell lines used were authenticated.* |
| Mycoplasma contamination | *Confirm that all cell lines tested negative for mycoplasma contamination OR describe the results of the testing for mycoplasma contamination OR declare that the cell lines were not tested for mycoplasma contamination.* |
| Commonly misidentified lines (See ICLAC register) | *Name any commonly misidentified cell lines used in the study and provide a rationale for their use.* |

## Palaeontology and Archaeology

| | |
|---|---|
| Specimen provenance | *Provide provenance information for specimens and describe permits that were obtained for the work (including the name of the issuing authority, the date of issue, and any identifying information). Permits should encompass collection and, where applicable, export.* |
| Specimen deposition | *Indicate where the specimens have been deposited to permit free access by other researchers.* |
| Dating methods | *If new dates are provided, describe how they were obtained (e.g. collection, storage, sample pretreatment and measurement), where they were obtained (i.e. lab name), the calibration program and the protocol for quality assurance OR state that no new dates are provided.* |

☐ Tick this box to confirm that the raw and calibrated dates are available in the paper or in Supplementary Information.

| | |
|---|---|
| Ethics oversight | *Identify the organization(s) that approved or provided guidance on the study protocol, OR state that no ethical approval or guidance was required and explain why not.* |

Note that full information on the approval of the study protocol must also be provided in the manuscript.

## Animals and other research organisms

Policy information about studies involving animals; ARRIVE guidelines recommended for reporting animal research, and Sex and Gender in Research

| | |
|---|---|
| Laboratory animals | *For laboratory animals, report species, strain and age OR state that the study did not involve laboratory animals.* |

| Wild animals | *Provide details on animals observed in or captured in the field; report species and age where possible. Describe how animals were caught and transported and what happened to captive animals after the study (if killed, explain why and describe method; if released, say where and when) OR state that the study did not involve wild animals.* |
|---|---|
| Reporting on sex | *Indicate if findings apply to only one sex; describe whether sex was considered in study design, methods used for assigning sex. Provide data disaggregated for sex where this information has been collected in the source data as appropriate; provide overall numbers in this Reporting Summary. Please state if this information has not been collected. Report sex-based analyses where performed, justify reasons for lack of sex-based analysis.* |
| Field-collected samples | *For laboratory work with field-collected samples, describe all relevant parameters such as housing, maintenance, temperature, photoperiod and end-of-experiment protocol OR state that the study did not involve samples collected from the field.* |
| Ethics oversight | *Identify the organization(s) that approved or provided guidance on the study protocol, OR state that no ethical approval or guidance was required and explain why not.* |

Note that full information on the approval of the study protocol must also be provided in the manuscript.

# Clinical data

Policy information about clinical studies
All manuscripts should comply with the ICMJE guidelines for publication of clinical research and a completed CONSORT checklist must be included with all submissions.

| Clinical trial registration | *Provide the trial registration number from ClinicalTrials.gov or an equivalent agency.* |
|---|---|
| Study protocol | *Note where the full trial protocol can be accessed OR if not available, explain why.* |
| Data collection | *Describe the settings and locales of data collection, noting the time periods of recruitment and data collection.* |
| Outcomes | *Describe how you pre-defined primary and secondary outcome measures and how you assessed these measures.* |

# Dual use research of concern

Policy information about dual use research of concern

## Hazards

Could the accidental, deliberate or reckless misuse of agents or technologies generated in the work, or the application of information presented in the manuscript, pose a threat to:

No | Yes
- [ ] [ ] Public health
- [ ] [ ] National security
- [ ] [ ] Crops and/or livestock
- [ ] [ ] Ecosystems
- [ ] [ ] Any other significant area

## Experiments of concern

Does the work involve any of these experiments of concern:

No | Yes
- [ ] [ ] Demonstrate how to render a vaccine ineffective
- [ ] [ ] Confer resistance to therapeutically useful antibiotics or antiviral agents
- [ ] [ ] Enhance the virulence of a pathogen or render a nonpathogen virulent
- [ ] [ ] Increase transmissibility of a pathogen
- [ ] [ ] Alter the host range of a pathogen
- [ ] [ ] Enable evasion of diagnostic/detection modalities
- [ ] [ ] Enable the weaponization of a biological agent or toxin
- [ ] [ ] Any other potentially harmful combination of experiments and agents

# Plants

Seed stocks

*Report on the source of all seed stocks or other plant material used. If applicable, state the seed stock centre and catalogue number. If plant specimens were collected from the field, describe the collection location, date and sampling procedures.*

Novel plant genotypes

*Describe the methods by which all novel plant genotypes were produced. This includes those generated by transgenic approaches, gene editing, chemical/radiation-based mutagenesis and hybridization. For transgenic lines, describe the transformation method, the number of independent lines analyzed and the generation upon which experiments were performed. For gene-edited lines, describe the editor used, the endogenous sequence targeted for editing, the targeting guide RNA sequence (if applicable) and how the editor was applied.*

Authentication

*Describe any authentication procedures for each seed stock used or novel genotype generated. Describe any experiments used to assess the effect of a mutation and, where applicable, how potential secondary effects (e.g. second site T-DNA insertions, mosiacism, off-target gene editing) were examined.*

# ChIP-seq

## Data deposition

☐ Confirm that both raw and final processed data have been deposited in a public database such as GEO.

☐ Confirm that you have deposited or provided access to graph files (e.g. BED files) for the called peaks.

Data access links
*May remain private before publication.*

*For "Initial submission" or "Revised version" documents, provide reviewer access links.  For your "Final submission" document, provide a link to the deposited data.*

Files in database submission

*Provide a list of all files available in the database submission.*

Genome browser session
(e.g. UCSC)

*Provide a link to an anonymized genome browser session for "Initial submission" and "Revised version" documents only, to enable peer review.  Write "no longer applicable" for "Final submission" documents.*

## Methodology

Replicates

*Describe the experimental replicates, specifying number, type and replicate agreement.*

Sequencing depth

*Describe the sequencing depth for each experiment, providing the total number of reads, uniquely mapped reads, length of reads and whether they were paired- or single-end.*

Antibodies

*Describe the antibodies used for the ChIP-seq experiments; as applicable, provide supplier name, catalog number, clone name, and lot number.*

Peak calling parameters

*Specify the command line program and parameters used for read mapping and peak calling, including the ChIP, control and index files used.*

Data quality

*Describe the methods used to ensure data quality in full detail, including how many peaks are at FDR 5% and above 5-fold enrichment.*

Software

*Describe the software used to collect and analyze the ChIP-seq data. For custom code that has been deposited into a community repository, provide accession details.*

# Flow Cytometry

## Plots

Confirm that:

☐ The axis labels state the marker and fluorochrome used (e.g. CD4-FITC).

☐ The axis scales are clearly visible. Include numbers along axes only for bottom left plot of group (a 'group' is an analysis of identical markers).

☐ All plots are contour plots with outliers or pseudocolor plots.

☐ A numerical value for number of cells or percentage (with statistics) is provided.

## Methodology

Sample preparation

*Describe the sample preparation, detailing the biological source of the cells and any tissue processing steps used.*

Instrument

*Identify the instrument used for data collection, specifying make and model number.*

Software

*Describe the software used to collect and analyze the flow cytometry data. For custom code that has been deposited into a community repository, provide accession details.*

| Cell population abundance | Describe the abundance of the relevant cell populations within post-sort fractions, providing details on the purity of the samples and how it was determined. |
| Gating strategy | Describe the gating strategy used for all relevant experiments, specifying the preliminary FSC/SSC gates of the starting cell population, indicating where boundaries between "positive" and "negative" staining cell populations are defined. |

☐ Tick this box to confirm that a figure exemplifying the gating strategy is provided in the Supplementary Information.

# Magnetic resonance imaging

## Experimental design

| Design type | Indicate task or resting state; event-related or block design. |
| Design specifications | Specify the number of blocks, trials or experimental units per session and/or subject, and specify the length of each trial or block (if trials are blocked) and interval between trials. |
| Behavioral performance measures | State number and/or type of variables recorded (e.g. correct button press, response time) and what statistics were used to establish that the subjects were performing the task as expected (e.g. mean, range, and/or standard deviation across subjects). |

## Acquisition

| Imaging type(s) | Specify: functional, structural, diffusion, perfusion. |
| Field strength | Specify in Tesla |
| Sequence & imaging parameters | Specify the pulse sequence type (gradient echo, spin echo, etc.), imaging type (EPI, spiral, etc.), field of view, matrix size, slice thickness, orientation and TE/TR/flip angle. |
| Area of acquisition | State whether a whole brain scan was used OR define the area of acquisition, describing how the region was determined. |

Diffusion MRI    ☐ Used    ☐ Not used

## Preprocessing

| Preprocessing software | Provide detail on software version and revision number and on specific parameters (model/functions, brain extraction, segmentation, smoothing kernel size, etc.). |
| Normalization | If data were normalized/standardized, describe the approach(es): specify linear or non-linear and define image types used for transformation OR indicate that data were not normalized and explain rationale for lack of normalization. |
| Normalization template | Describe the template used for normalization/transformation, specifying subject space or group standardized space (e.g. original Talairach, MNI305, ICBM152) OR indicate that the data were not normalized. |
| Noise and artifact removal | Describe your procedure(s) for artifact and structured noise removal, specifying motion parameters, tissue signals and physiological signals (heart rate, respiration). |
| Volume censoring | Define your software and/or method and criteria for volume censoring, and state the extent of such censoring. |

## Statistical modeling & inference

| Model type and settings | Specify type (mass univariate, multivariate, RSA, predictive, etc.) and describe essential details of the model at the first and second levels (e.g. fixed, random or mixed effects; drift or auto-correlation). |
| Effect(s) tested | Define precise effect in terms of the task or stimulus conditions instead of psychological concepts and indicate whether ANOVA or factorial designs were used. |

Specify type of analysis:    ☐ Whole brain    ☐ ROI-based    ☐ Both

Statistic type for inference    Specify voxel-wise or cluster-wise and report all relevant parameters for cluster-wise methods.

(See Eklund et al. 2016)

| Correction | Describe the type of correction and how it is obtained for multiple comparisons (e.g. FWE, FDR, permutation or Monte Carlo). |

## Models & analysis

nature portfolio | reporting summary

| n/a | Involved in the study |
|---|---|
| ☐ | ☐ Functional and/or effective connectivity |
| ☐ | ☐ Graph analysis |
| ☐ | ☐ Multivariate modeling or predictive analysis |

**Functional and/or effective connectivity**

*Report the measures of dependence used and the model details (e.g. Pearson correlation, partial correlation, mutual information).*

**Graph analysis**

*Report the dependent variable and connectivity measure, specifying weighted graph or binarized graph, subject- or group-level, and the global and/or node summaries used (e.g. clustering coefficient, efficiency, etc.).*

**Multivariate modeling and predictive analysis**

*Specify independent variables, features extraction and dimension reduction, model, training and evaluation metrics.*

