## [Peer Review File · Nature Methods]

Peer Review Information

Manuscript Title: Analyzing single-cell bisulfite sequencing data with 'MethSCAn'

Corresponding author name(s): Lukas P. M. Kremer, Ana Martin-Villalba, Simon Anders

Editorial Notes:

Reviewer Comments & Decisions:

Decision Letter, initial version:
--

7th Jul 2023

Dear Dr Anders,

Your Article, "Analyzing single-cell bisulfite sequencing data with "scbs"", has now been seen by 3 reviewers. As you will see from their comments below, although the reviewers find your work of considerable potential interest, they have raised a number of concerns. We are interested in the possibility of publishing your paper in Nature Methods, but would like to consider your response to these concerns before we reach a final decision on publication.

We therefore invite you to revise your manuscript to address these concerns. In particular, we ask you to perform more benchmarking comparisons against existing tools as well as using datasets suggested by the reviewers.

* include a point-by-point response to the reviewers and to any editorial suggestions

* please underline/highlight any additions to the text or areas with other significant changes to facilitate review of the revised manuscript

- * address the points listed described below to conform to our open science requirements
- * ensure it complies with our general format requirements as set out in our guide to authors at www.nature.com/naturemethods
- * resubmit all the necessary files electronically by using the link below to access your home page

[REDACTED]

We hope to receive your revised paper within 12 weeks. If you cannot send it within this time, please let us know. In this event, we will still be happy to reconsider your paper at a later date so long as nothing similar has been accepted for publication at Nature Methods or published elsewhere.

OPEN SCIENCE REQUIREMENTS

REPORTING SUMMARY AND EDITORIAL POLICY CHECKLISTS

IMAGE INTEGRITY

When submitting the revised version of your manuscript, please pay close attention to our Digital Image Integrity Guidelines and to the following points below:

-- that unprocessed scans are clearly labelled and match the gels and western blots presented in

figures.

- that control panels for gels and western blots are appropriately described as loading on sample processing controls
- all images in the paper are checked for duplication of panels and for splicing of gel lanes.

DATA AVAILABILITY

All novel DNA and RNA sequencing data, protein sequences, genetic polymorphisms, linked genotype and phenotype data, gene expression data, macromolecular structures, and proteomics data must be deposited in a publicly accessible database, and accession codes and associated hyperlinks must be provided in the "Data Availability" section.

CODE AVAILABILITY

Please include a "Code Availability" subsection in the Online Methods which details how your custom code is made available. Only in rare cases (where code is not central to the main conclusions of the

paper) is the statement “available upon request” allowed (and reasons should be specified).

For more information on our code sharing policy and requirements, please see:
<https://www.nature.com/nature-research/editorial-policies/reporting-standards#availability-of-computer-code>

ORCID

Nature Methods is committed to improving transparency in authorship. As part of our efforts in this direction, we are now requesting that all authors identified as ‘corresponding author’ on published papers create and link their Open Researcher and Contributor Identifier (ORCID) with their account on the Manuscript Tracking System (MTS), prior to acceptance. This applies to primary research papers only. ORCID helps the scientific community achieve unambiguous attribution of all scholarly contributions. You can create and link your ORCID from the home page of the MTS by clicking on ‘Modify my Springer Nature account’. For more information please visit please visit www.springernature.com/orcid.

Sincerely,
Lei

Lei Tang, Ph.D.
Senior Editor
Nature Methods

Reviewers' Comments:

Reviewer #1:
Remarks to the Author:

Kremer and colleagues have proposed an innovative dimensionality reduction technique, specifically tailored for the analysis of single-cell bisulfite sequencing data. This novel approach is predicated on the understanding that fixed-size window tiling can result in the dilution of cell type-specific signals. As an alternative, the authors have suggested a process whereby variably methylated regions (VMRs) are identified first, followed by the execution of dimensionality reduction on these VMRs. The study compellingly demonstrates that this novel approach can yield improved discrimination of cell types.

Although the limitations of a fixed size window average have long been suspected, no previous studies

have explored this issue with the depth and rigor seen in the current work. The authors compellingly demonstrated the superiority of the VMR-based method over the conventional 100kb bin average.

However, a few points for consideration arise:

There are several existing methods that handle single-cell methylome dimensionality reduction and imputation, such as LIGER, scAI, EpiScanpy MOFA/MOFA+, which are general-purpose tools that support DNA methylation analysis, as well as others like DeepCpG, epiclomal, MELISSA and scMET that are specifically focused on DNA methylation (see PMID 35718270 for a recent review). The present manuscript, however, only appears to compare the new method with the simplest fixed-size window approach, without discussing these other, more recently developed methods.

It's not entirely clear how this method would be applicable to datasets where the primary source of biological variation isn't cell type but rather other biological factors such as normal vs. cancer, age, or different cell cycle stages. These factors could potentially lead to VMRs being present at various genomic scales. Are parameter adjustments necessary under these conditions?

While the direct detection of VMRs from data is certainly a desirable feature, it's worth noting that this approach could be computationally demanding. A comparison with a less complex alternative that aggregates methylation using known biological annotations of regulatory elements (for example, from ENCODE, as implemented in EpiScanpy) could provide additional insight into the relative advantages and disadvantages of the proposed method.

The method's effectiveness when applied to non-CG methylation should be explored, given that this form of methylation is recognized for its greater discriminatory power among neurons.

Philosophically, how does the method address the balance of information and sparsity? Smaller VMRs could indeed provide higher discriminatory power, but they may also be more affected by data sparsity, making imputation necessary for cross-cell comparison.

It seems the method still relies on choosing a small window size and a step size for VMR detection. How would this parameter be chosen and how does this affect the cell clustering?

The methodology was benchmarked against the authors' mouse brain data. It would be valuable to extend this comparison to other datasets, such as those produced by the research groups led by Joe Ecker and Wolf Reik.

In reference to Figure 4B and C, the utilization of neighbor scores to evaluate separation seems ambiguous. Lower neighbor scores can still result in clear separability, making the choice of these scores for analysis somewhat counterintuitive. Furthermore, the choice to perform this neighbor score analysis on PC space rather than the UMAP as depicted in Figures B and C could use further clarification.

Has the team attempted to use tSNE with varying perplexity parameters? This could be another beneficial comparison to further evaluate the effectiveness of their proposed method.

Reviewer #2:

Remarks to the Author:

Kremer et al have presented a paper describing a novel approach to the quantitation, normalisation and analysis of single cell bisulphite data which aims to address issues of technical bias and signal dilution in some of the existing approaches. They provide a description of their method, a python software package to implement the processing steps (supplemented by example R code for some parts of the described analysis), and an example analysis of data from both their group and an external dataset.

The paper is well written, with a clear description of the problems with current approaches, and justifications for their new methods. The method itself is clearly and concisely described. I was able to install their software from the PyPi repository and could follow through all the steps in their analysis using the example code on their site.

In general this method is an interesting and well thought out approach which offers improvements over existing methods, however there are some places where additional clarification or illustration would be useful.

The initial step in the analysis turns the measured methylation values into residuals to a globally calculated methylation value across all cells. The initial description of this in the paper talks about subtracting the mean methylation for each position, but if I'm reading it correctly it's actually the smoothed running value which is used.

[1] It would be useful to have some comment on the density of data which would be necessary for this approach. The datasets used by the authors feature thousands of cells, but many scBS experiments are much smaller than that, and in those a large proportion of all CpG positions will be measured in only 1 or 2 cells, giving little opportunity for calculating sensible global values against which to normalise.

[2] In a few places in the paper (kernel smoothing and detection of variable and differentially methylated regions) the methods use small, fixed size windows, as the basis for analysis - often 1000bp. This size of window can be problematic in BS-Seq data given the uneven distribution of CpGs across the genome, and the potential for this to introduce bias into the results. Would smoothing in windows of fixed numbers of CpGs make more sense?

[3] When selecting the variably methylated regions the method takes the mean of the residuals in a given region. Does this mean that if adjacent regions changed in opposite directions that this would be averaged away and the region discarded as uninteresting?

The authors illustrate the effectiveness of their method by showing a PCA of a set of cells separated by their VMR process, compared to a more conventional separation by simple methylation calculation. In the VMR PCA there are two potentially influential steps - the conversion of methylation to residuals, and the imputation of missing values by iterative PCA. It would be useful to know which of these was having the greatest influence, since it's possible that it is the imputation which is causing the clearer separation through the reinforcement of the initial signal, rather than that the initial values are clearer.

[4] What would a PCA of the VMR values with missing values still set to 0 look like? This would be a

more direct comparison to the global calculation and would demonstrate which step is more important in the improved separation they show.

[5] In the DMR detection they show a comparison of 130 cells and 58 cells, but am I right in thinking that these cells were split off only after calculating the normalised VMR values from a much larger population of cells? Does the approach still work as effectively if starting from the 188 cells in the DMR analysis, or do you require the context of the larger dataset to get accurate VMR values to put into the DMR detection?

[6] In the DMR calculation the false discovery rate is estimated by shuffling cell labels and repeating the analysis - which seems to be a reasonable idea for this type of data. However, the FDR estimation is done only after the initially calculated windows are filtered for the top 2% then merged if they are adjacent. The values tested are those from the recalculated t-statistic after merging. I would assume that the approach shown heavily favours regions which are much larger than the initial window size, and that the major contribution to significance is adjacency in the initially filtered top 2% of regions. Does this method work if the FDR values are calculated from the original window data, before merging adjacent windows? If not, then what size of DMR is realistically detectable with this method?

Reviewer #3:

Remarks to the Author:

Kremer et al describe 'scbs' – a computational toolset for the analysis of single-cell methylation data. In the field there is certainly a need for a suite of tools to handle single-cell methylation data, and the manuscript details several compelling strategies and frameworks for doing so. The most significant aspect is the identification of variable regions that are then used for subsequent dimensionality reduction, clustering and visualization; followed by tools for DMR calling. The authors apply 'scbs' to their own, published, scNMT-seq dataset on brain as well as some applications to a dataset produced by the Ecker Lab, also brain. While there are positives to the work, there are key limitations in the assessment of the tool that need to be performed in order to properly evaluate the tool and its broader utility. Lastly, the name itself is a term that is already used to define an experimental approach "single-cell bisulfite sequencing" – whereas this reports on a computational tool for that type of data (or enzymatic-converted data, which is not bisulfite). I strongly suggest changing the name of the tool to something informative – e.g. SCMtools for "single-cell methylome tools" – anything that indicates that it is a computational tool and used for analysis of single-cell methylome data.

Major Comments:

The authors perform almost all of their evaluation solely on their own dataset with very little analysis of the Luo et al 2017 dataset. Furthermore, their only comparison of methodology is by using 100 kbp tiling windows for CpG methylation levels, which does not seem all that appropriate. Using relevant windows – e.g. all annotated promoters and enhancers like geneHancer or something like that is a much more reasonable strategy. Even just all promoter regions such as TSS +/- 2 kbp or something.

They do not compare cluster resolution with CH methylation windows which is generally used for neuron subtype clustering (as often CpG methylation will be the same between two subtypes of the same class, eg excitatory, but the CH levels will be different). Does the tool work with CH

methylation? How does the variable window approach for CpG methylation look compared to 100 kbp tiling CH methylation (where 100 kbp tiles make sense due to the structure of the mark in neurons).

These analyses should be carried out on both their own dataset, the Luo et al dataset, but also on datasets from other tissue types. For example:

Argelaguet R, Clark SJ, Mohammed H, Stapel LC, Krueger C, Kapourani CA, Imaz-Rosshandler I, Lohoff T, Xiang Y, Hanna CW, Smallwood S, Ibarra-Soria X, Buettner F, Sanguinetti G, Xie W, Krueger F, Göttgens B, Rugg-Gunn PJ, Kelsey G, Dean W, Nichols J, Stegle O, Marioni JC, Reik W. Multi-omics profiling of mouse gastrulation at single-cell resolution. *Nature*. 2019 Dec;576(7787):487-491. doi: 10.1038/s41586-019-1825-8. Epub 2019 Dec 11.

Chatterton, Z., Lamichhane, P., Ahmadi Rastegar, D. et al. Single-cell DNA methylation sequencing by combinatorial indexing and enzymatic DNA methylation conversion. *Cell Biosci* 13, 2 (2023). <https://doi.org/10.1186/s13578-022-00938-9> (Enzymatic conversion workflow)

Furthermore, the tool is demonstrated on relatively small datasets on the order of ~3000 cells; however, single-cell methylation datasets are increasing in size – the Luo et al dataset used is quite old at this point, with much larger datasets produced by the Ecker and Luo labs, with hundreds-of-thousands of cells. It is likely that such datasets will become more common and any tool that will reach wide adoption will have to be able to handle large datasets. At least one of these should be used as an example with compute times reported – eg:

Liu, H., Zhou, J., Tian, W. et al. DNA methylation atlas of the mouse brain at single-cell resolution. *Nature* 598, 120–128 (2021). <https://doi.org/10.1038/s41586-020-03182-8>

(~100k cells)

Or the newer dataset that is ~300-400k cells; however, the 100k cell dataset should be sufficient for demonstration purposes. Again comparing CH methylation 100 kbp windows (standard for neurons) vs variable windows vs promoters & enhancers.

Author Rebuttal to Initial comments

Response to Reviewers

We thank all Reviewers for their thorough and helpful analysis of our work. We discuss all their concerns in the following.

In the revised manuscript, all changed or new text is highlighted by **blue text color**. In this response, quote from the reviewer's reports are set in **teal color**.

Reviewer #1:

Remarks to the Author: Kremer and colleagues have proposed an innovative dimensionality reduction technique, specifically tailored for the analysis of single-cell bisulfite sequencing data. This novel approach is predicated on the understanding that fixed-size window tiling can result in the dilution of cell type-specific signals. As an alternative, the authors have suggested a process whereby variably methylated regions (VMRs) are identified first, followed by the execution of dimensionality reduction on these VMRs. The study compellingly demonstrates that this novel approach can yield improved discrimination of cell types.

Although the limitations of a fixed size window average have long been suspected, no previous studies have explored this issue with the depth and rigor seen in the current work. The authors compellingly demonstrated the superiority of the VMR-based method over the conventional 100kb bin average.

We thank the Reviewer for this clear endorsement of the value of our work.

However, a few points for consideration arise:

There are several existing methods that handle single-cell methylome dimensionality reduction and imputation, such as LIGER, scAI, EpiScanpy MOFA/MOFA+, which are general-purpose tools that support DNA methylation analysis, as well as others like DeepCpG, epiclomal, MELISSA and scMET that are specifically focused on DNA methylation (see PMID 35718270 for a recent review). The present manuscript, however, only appears to compare the new method with the simplest fixed-size window approach, without discussing these other, more recently developed methods.

We agree that it is important to place our work into the context of existing methods, and we have now expanded the Discussion section in this regard.

In this context, we should reiterate that a core strength of our method is that it produces data matrices that can be analysed with methods for single-cell *RNA-Seq* data: scbs produces a features-by-cells matrix that can be treated in the same manner as log-normalized gene-by-cells expression matrices. We should therefore ask whether the methods the Reviewer mention offer alternative ways to prepare such matrices or to process them.

- MOFA/MOFA+ (multi-omics factor analysis) expects feature-by-cell matrices as input. It is a method for integration of multi-modal data, i.e., it takes several matrices from different modalities (say, gene expression, methylation and more) and combines them into a single, integrated matrix,

which can then be further processed with standard approaches for dimension reduction, clustering, etc. If a user had multi-omics data, e.g., data from single-cell NMT-Seq, which comprises, for each cell, data on gene expression, DNA methylation and DNA accessibility, then scbs could be used to prepare the feature-by-gene matrix for the DNA methylation (and, in fact, also for the accessibility), while standard counting followed by log-normalization is used for the gene expression matrix. Then, MOFA+ may be used downstream of scbs, to integrate the three matrices into one. Therefore, MOFA+ is not an alternative to scbs, but rather a tool that can be combined with scbs in case of multi-omics data. In fact, we used this approach in our multi-omics study described in Kremer, Cerrizuela et al., bioRxiv, 2022 (10.1101/2022.07.13.499860).

- However, in one specific detail, MOFA/MOFA+ can, in fact, be seen as an alternative to one of the steps in scbs: The first step of standard downstream analysis is to reduce the dimension of the matrix columns from the tens of thousands of features to a few dozen latent variables, and this is usually done by principal component analysis (PCA). As standard PCA cannot handle the missing values that are unavoidable in scBS data due to limited coverage, we use a modified, “iterative”, PCA, as described in our Methods section. Replacing PCA with factor analysis (FA) seems like another viable way of treating the missing values, as FA, especially when implemented in Bayesian fashion (as is done in MOFA), can handle missing values in a natural way. Therefore, the Reviewer’s comment made us wonder whether results would be further improved if we replaced our iterative PCA by MOFA’s factor analysis, i.e., using MOFA not for multi-omics but for single-omics factor analysis. We have therefore expanded our benchmarks and now also test how results change when the iterative PCA is replaced by MOFA’s FA implementation. The outcome was quite mixed, however: it seems that in most cases, MOFA-style FA did not yield better results than our iterative PCA. (This should not be considered as a negative for MOFA, though: After all, MOFA is meant to be used on multi-omics data, and as we are here only discussing a single modality, MOFA cannot show its core strengths.)
- LIGER is meant for similar tasks as MOFA/MOFA+, namely also to integrate data from different modalities. Therefore, the same applies as stated above for MOFA/MOFA+: If one has several modalities, it may be useful to use scbs to prepare the feature-by-gene matrix for the methylation data, which one then hands to LIGER, together with the matrices from the other modalities, for integration. In our Discussion, we therefore now state that matrices produced with scbs are suitable as input to multi-omics integration methods such as MOFA/MOFA+ or LIGER.

We also would like to comment on the other tools that the Reviewer mentioned:

- EpiScanpy is an extension to the well-known Scanpy toolkit. The core strength of EpiScanpy is that these downstream analysis and visualization methods are offered in an interactive, user-friendly interface that promotes exploratory data analysis. Its functionality partially overlaps with that of scbs: Given a list of genomic regions EpiScanpy produces a features-by-cells matrix that is suitable for analysis with scRNA-seq methods. However, one core functionality of scbs (VMR detection) lies upstream of this step, while another core functionality (quantifying methylation using the shrunken mean of residuals) is an alternative strategy for this step. In contrast, EpiScanpy is more

tailored towards downstream analysis and interactive data exploration. Therefore scbs and EpiScanpy are not competitors, but rather complementary methods that can be used in a joint analysis workflow: A VMR-by-cells matrix generated with scbs can be explored and visualized with EpiScanpy. We now point this out in the Discussion.

- scAI is another integration method, this time specifically to integrate single-cell ATAC-Seq with RNA-Seq data. There is a subtle but very important difference between transposase-based methods (ATAC-Seq) and bisulfite-based (as in scNMT-Seq) methods, namely that for the former, we either see a read for a region or not and use this as our observed data, while for the latter, we only consider regions covered by reads as observed and check there whether we the reads show base conversions or not. So, in ATAC-Seq, we cannot truly distinguish non-accessible from non-observed regions, while our method leverages the fact that one can do that is scBS data. Moreover, scAI requires transcriptome data to work, while scbs is designed to rely only on the methylation data. Due to these fundamental differences, we feel that discussing tools for scATAC data is not helpful in a paper on scBS data analysis, and hope the Reviewer will not mind that we skip over scAI.
- The scMet tool sounds interesting, although it does a job somewhat different to what our tool does: Rather than searching the entire genome for variable regions, it simply identifies the most variable regions among a set of pre-defined, user-specified set of genomic intervals. We tried nevertheless to run it on our data, but unfortunately, the tool repeatedly reported “internal errors” when we tried to run its default variational Bayes mode. (The tool’s manual notes “For small datasets, we recommend using MCMC implementation since it is more stable.”, i.e., the authors seem to be aware that the tool has problems with larger data sets.) After several attempts with different sets of input data and genomic intervals, the tool only rarely ran to completion. Thus, we were unable to discuss scMet in more detail, but we have added a citation to the Discussion.
- We had similar problems with MELISSA: this tool, published in 2019, seems to have been written at a time when cell numbers in scBS data were much lower. As it loads all data into RAM, it tends to have trouble with larger datasets. Nonetheless, we let this tool run on our data for a week, but unfortunately there was no indication of progress, and thus no results. We feel that one should consider this tool as outdated, and comparing with it would not be productive. (The scMet tool seems to be the successor to scMet, as it has the same first author.) We hope the Reviewer will forgive our skipping a detailed discussion of this tool. We skip it also as it is meant to impute missing values, and we feel that there is little gained in adding to the already large debate on merits vs dangers of imputation in single-cell data. However, we do cite it now as an early prior work in the Discussion.
- DeepCpG is another tool for imputing missing values. Here, we feel that its aim is too different from ours for a useful comparison, and we have therefore decided to not elaborate on it.

- epiclomal, finally, is a very specialized tool, namely for analysing within-tumour phylogeny — an important topic, but far from our aim, and hence a bit far from the paper’s scope.

In summary, our software `scbs` offers several novel methods that are not implemented in any of the abovementioned tools, namely:

- the ability to discover informative genomic intervals in the data itself, instead of relying on a user-specified set of features
- a new measure of DNA methylation in genomic intervals that accounts for biases introduced by the random positioning of reads
- the detection of differentially methylated regions in single-cell methylation data

As demonstrated in our recent study (Kremer, Cerrizuela et al., bioRxiv, 2022, 10.1101/2022.07.13.499860) our tool’s ability to discover epigenetic change outside previously annotated genomic regions can reveal new biology. We have added a section to the discussion in which we discuss `scbs` in the context of other tools.

Furthermore, to demonstrate our tool’s compatibility with dimensionality reduction tools, we included an analysis workflow in our benchmarks that uses `scbs` for constructing the feature-by-cell matrix and MOFA+ for dimensionality reduction.

It’s not entirely clear how this method would be applicable to datasets where the primary source of biological variation isn’t cell type but rather other biological factors such as normal vs. cancer, age, or different cell cycle stages. These factors could potentially lead to VMRs being present at various genomic scales. Are parameter adjustments necessary under these conditions?

We agree with the Reviewer that we should demonstrate our method with a sufficiently diverse selection of data sets. We have therefore added two more data sets and expanded our benchmark (see new Supplementary Figure S1, pasted below). To address the Reviewer’s specific concern about variation due to factors other than cell type, we added the dataset by Argelaguet et al. (2019), who studied gastrulation in mouse embryos (embryonic days E4.5 to E7.5). We now show that our method works well to recapitulate the differences between embryonic developmental stage, the branching of the developing early embryo in the three germ layers, and the subsequent early branchings covered by that data set. To demonstrate our tool’s ability to dissect another biological feature that is not cell type, we furthermore included human colorectal cancer data from Bian et al. 2018. This data set comprises cells obtained from different tissue samples, namely primary tumor, normal adjacent tissue, lymph node metastases, liver metastases before and after treatment, and omental metastases. The result of the extended benchmark is now shown in the first row of our new Supplementary Figure S1:

We also provide further evidence for our claim that scbs is very robust to parameter change, i.e. the results do not strongly depend on the exact selection of parameters: As Supplementary Figure 2 (see below) now shows for two of the data sets, performance does depend only very weakly on the precise choices for the two main parameters, the bandwidth of the sliding window and the variance threshold. Specifically, we found that varying bandwidth from 500 bp to 5 kb does not affect performance, and only at 10 kb, we see the effect of oversmoothing.

A Kremer *et al.* dataset

B Luo *et al.* dataset

We agree with the Reviewer that different biological processes might affect methylation at different genomic length scales. The gastrulation process studied by Argelaguet *et al.* is a good example here: Early embryogenesis starts with a global removal of all DNA methylation, with the DNA being fully unmethylated at the morula stage (E3), and remethylation occurring on the following days. The data by Argelaguet *et al.* captures this global remethylation in its early time points and then the more region-specific processes in the later time points. Despite this change in scale, our method worked well for this data set (Supplementary Fig. S1D-E, see below). We hope that this satisfied the Reviewer's concern in this regard.

While the direct detection of VMRs from data is certainly a desirable feature, it's worth noting that this approach could be computationally demanding. A comparison with a less complex alternative that aggregates methylation using known biological annotations of regulatory elements (for example, from ENCODE, as implemented in EpiScanpy) could provide additional insight into the relative advantages and disadvantages of the proposed method.

We thank the Reviewer for raising this important point. While we have argued that most existing work uses simple tiling of the genome into equal-sized windows, we have in fact neglected to discuss the other simple but important alternative that the Reviewer mentions here. Therefore, we have substantially expanded our benchmark to include a comparison of our VMRs with ENCODE regulatory elements. The results were insightful, as the Reviewer expected.

Our revised benchmark (Figure 4D and Supplementary Figure 1, see below) now compares four sources of intervals: simple genomic tiling in 100 kb bins (yellow lines), all promoter regions (green), the ENCODE candidate cis-regulatory elements (cCREs) as suggested by the Reviewer (red), and our VMRs (green). The performance of the cCREs was, in fact, comparable to that of the VMRs.

However, there are ~340k cCRE features in the current ENCODE annotation data set for the mouse, while our VMR analysis used only ~60k features. Hence, while the Reviewer expected the use of annotated regulatory regions to be less computationally demanding than inferring VMRs, the opposite turned out to be true: the dimension reduction via PCA for the large amount of cCREs drove the runtime up to over 2 hours, while the process took only a few minutes for the VMRs, with about half the time used for inferring the VMRs and the other for performing the PCA. (This is due to the fact that PCA scales quadratic in the number of regions while VMR detection does not.) Of course, the larger matrix also required much more RAM:

Of course, due to the difference in number of regions, this is not a fair comparison. We therefore performed feature selection on the cCREs by retaining only those 60k cCREs that had highest read coverage in the data, which, as expected, reduced runtime and memory needs to the same

level as with VMRs (Supplementary Figure S3C-D, see below). However, with the reduced cCRE set (light red), cell type distinction was now noticeably compromised (Supplementary Figure S3A-B).

We conclude from this that using methylation at VMRs provides a more information-dense picture of cell state than methylation at annotated cCREs. It could be that this is simply because the Encode annotation is not well-suited for the neuronal cell types studied here but it might also indicate that studying DNA methylation can detect regions of regulatory importance that cannot be seen well by the approaches used by Encode. In any case, it underpins the value of having a method for unbiased *de novo* detection of regions of varying DNA methylation. Of note, this approach also works for non-model organisms, where regulatory annotations are not available.

The method's effectiveness when applied to non-CG methylation should be explored, given that this form of methylation is recognized for its greater discriminatory power among neurons.

We think that this is a very reasonable request and agree that its important to showcase our methods on CH methylation data.

Thus, we have re-analyzed the Luo et al. data set using three different strategies:

- the classic approach used by Luo et al.: averaging CH methylation in 100 kb genomic tiles
- applying our approach on CH methylation data: using the CH data for VMR detection, quantifying CH methylation in these intervals with the shrunken mean of the residuals
- applying our approach on CpG methylation data, as was also done in our benchmarks.

Due to the lack of ground-truth cell-type labels that are independent of the CH/CpG data, it was not possible to quantify cell type separation as done in our benchmarks. Nonetheless, visual inspection of the three resulting UMAPs clearly showed that all three approaches were able to separate the neuronal subtypes that Luo et al. annotated based on CH methylation in 100 kb genomic tiles (Supplementary Figure S4A, see below). This analysis demonstrates that our tool also works with CH methylation data.

Philosophically, how does the method address the balance of information and sparsity?
Smaller VMRs could indeed provide higher discriminatory power, but they may also be more affected by data sparsity, making imputation necessary for cross-cell comparison.

As outlined by the reviewer, it is not intuitively clear whether large or small genomic intervals are more suitable for the analysis of single-cell methylation data. On the one hand, large genomic intervals offer the clear advantage that they reduce the proportion of missing values in the resulting feature-by-cell matrix, as it becomes very unlikely that a very large interval does not contain a sequencing read. Thus, large intervals seem like a natural choice. On the other hand, it is also clear that the signal present in the methylation data becomes increasingly “diluted” as larger and larger intervals are selected. For this reason, small precise intervals seem like the more appropriate choice. Both options clearly have their appeal and to us it was not immediately obvious which option is preferable, which is why we performed extensive benchmarks using

various analysis strategies reported in this manuscript. As it turns out, despite introducing additional missing values, our strategy of choosing small, informative intervals outperforms the commonly used strategy where the genome is divided into 100 kb tiles.

It seems the method still relies on choosing a small window size and a step size for VMR detection. How would this parameter be chosen and how does this affect the cell clustering?

To address this valid question, we performed an additional parameter sweep of the step size parameter. This benchmark revealed that step sizes between 5bp and 200bp perform very similarly, while performance dropped when using a step size of 500bp or 1000bp (Supplementary Fig. S2C).

As a greater step size leads to faster runtime of the VMR detection algorithm, we thus increased the default step size from 10bp to 100bp. We thank the reviewer for their valuable suggestion.

The methodology was benchmarked against the authors' mouse brain data. It would be valuable to extend this comparison to other datasets, such as those produced by the research groups led by Joe Ecker and Wolf Reik.

We agree that the inclusion of more data sets would strengthen our benchmark. The challenge lies in finding a data set with cell-type labels that are suitable to be taken as ground truth, i.e. cell type labels derived not from methylation data but from transcriptomic data. This rules out pure methylation data sets, while single-cell multi-omic data sets (e.g. scNMT-seq) work. Therefore, we followed the Reviewer’s suggestion to try data from Wolf Reik’s lab and have now included their scNMT gastrulation data set into our benchmarks (Supplementary Fig. S1D-E).

Furthermore, we also demonstrated our tool on a human colorectal cancer data set in which we use the single-cell methylome data to distinguish cells sampled from healthy and malignant tissue (Bian et al. 2018, Supplementary Fig S1C). Finally, we also performed a “stress test” on a very large 100k-cell neural data set (Liu et al. 2021, Supplementary Fig S4B):

B Liu *et al.* data set (~100,000 cells)
CpG, shrunken mean of residuals in VMRs

In reference to Figure 4B and C, the utilization of neighbor scores to evaluate separation seems ambiguous. Lower neighbor scores can still result in clear separability, making the choice of these scores for analysis somewhat counterintuitive. Furthermore, the choice to perform this

neighbor score analysis on PC space rather than the UMAP as depicted in Figures B and C could use further clarification.

It is actually essential to perform the neighbor score analysis in PC space and not in UMAP space. The reason is that we want to evaluate the quality or information content of the processed data with regards to how suitable it is for typical further downstream analysis tasks like clustering or trajectory inference. Both of these steps rely on the so-called kNN graph, which connects each cell with its k nearest neighbors. In modularity clustering with the Leiden algorithm (a standard approach in single-cell analysis), clusters are then determined as groups of cells that are highly connected with each other in the kNN graph. Similarly, in typical trajectory inference methods such as diffusion maps, trajectories follow strong paths in the graphs. In all these applications, it is crucial that the kNN graph is formed by quantifying similarity between putative neighboring cells in the PCA space (or in another intermediate space of similar dimensionality). This is because this space has been constructed with the goal of getting optimal signal-to-noise ratio. In contrast, the UMAP space is constructed with the goal of being two-dimensional, as a 2D visualization is sought. In the early days of single-cell analysis, it was a common pitfall to perform inference such as judging cell-to-cell similarity by inspecting the UMAP directly rather than going back to PCA space. However, studies dedicated to this topic have since pointed out why this is bad practice (e.g., <https://doi.org/10.1371/journal.pcbi.1011288>). As our benchmark aims to score the suitability of the produced data matrix for the mentioned downstream task (which widely rely on the kNN graph), it makes sense to check whether neighborhood relationships (and hence the kNN graph) show good overlap with what the ground-truth annotation shows, and therefore, we have to assess neighborhood in the same way as it is done when constructing the kNN graph or other input data for the downstream analysis methods typically used in single-cell analysis — and that is Euclidean distance in PCA space.

We hope that clears up the Reviewer's question.

We also note that for the mentioned reason, the 2D UMAP does not always give a clear indication on how separable two cell populations really are in PCA space, which might be why the reviewer feels that sometimes the separation score does not agree with what is shown in the UMAP.

Has the team attempted to use tSNE with varying perplexity parameters? This could be another beneficial comparison to further evaluate the effectiveness of their proposed method.

As just argued, the separation in the 2D visualization is not informative as it is the separation in PCA space that is important. The UMAPs are only provided as an informal visual indication of data quality that cannot replace the formal analysis with PCA-based neighbor score but is included as a visual supplement to it — and in fact, this is how 2D visualizations, both UMAP

and t-SNE, should always only be used in single-cell analysis, as has been pointed out by the paper just cited and many other authors. Therefore, we feel that playing with UMAP or tSNE hyperparameters would be misleading as it would seem to support the wrong practice of basing analysis on distorted 2D visualizations.

Reviewer #2:

Remarks to the Author: Kremer et al have presented a paper describing a novel approach to the quantitation, normalisation and analysis of single cell bisulphite data which aims to address issues of technical bias and signal dilution in some of the existing approaches. They provide a description of their method, a python software package to implement the processing steps (supplemented by example R code for some parts of the described analysis), and an example analysis of data from both their group and an external dataset.

The paper is well written, with a clear description of the problems with current approaches, and justifications for their new methods. The method itself is clearly and concisely described. I was able to install their software from the PyPi repository and could follow through all the steps in their analysis using the example code on their site.

We thank the Reviewer for the endorsement of our work and of the quality of our software.

In general this method is an interesting and well thought out approach which offers improvements over existing methods, however there are some places where additional clarification or illustration would be useful.

The initial step in the analysis turns the measured methylation values into residuals to a globally calculated methylation value across all cells. The initial description of this in the paper talks about subtracting the mean methylation for each position, but if I'm reading it correctly it's actually the smoothed running value which is used.

We thank the Reviewer for catching this potential source of misunderstandings in our description of the method. The residuals are indeed calculated by subtracting the *smoothed* global methylation average of a given position, and not by simply subtracting the mean. In our previous iteration of the manuscript, we first introduced the concept of calculating the residuals by subtracting the mean and only later explained that this mean is obtained using a kernel smoother. To prevent misunderstandings, we now briefly mention early on that the average is a smoothed average, and later elaborate how the smoothing is achieved.

[1] It would be useful to have some comment on the density of data which would be necessary for this approach. The datasets used by the authors feature thousands of cells, but many scBS experiments are much smaller than that, and in those a large proportion of all CpG positions will be measured in only 1 or 2 cells, giving little opportunity for calculating sensible global values against which to normalise.

We fully agree with the statement that small single-cell methylome data sets are much more difficult to analyze. This is why our benchmarks include random sub-samples of each data set, which allowed us to explore how our methods fare on smaller data sets. The smallest of these samples comprise only 100 cells. To ensure that our benchmark analysis of these sub-sampled data sets is fair, we made sure to run the entire analysis from start to finish only on the reduced set of cells. This means that both the smoothed global averages were computed not on the full data, but rather re-computed on the smaller data sets. Similarly, VMRs were re-discovered for each sub-sampled data set. As the reviewer suspected, the ability to distinguish cell types (as measured by the neighbor score) sharply drops in small data sets (Figure 4D):

As part of the revisions, we also repeated our benchmarks on three other data sets (Supplementary Figure S1):

The trend that cell groups are harder to distinguish in smaller data sets is clearly observed across all combinations of analysis methods and in all four data sets. For all data sets in our benchmarks, the neighbor score starts to plateau starting at roughly the 1000-cell mark. On the one hand, we thus feel very tempted to report this cell number as the minimum target cell number for future single-cell methylome studies in the manuscript. On the other hand, it seems likely that other factors such as the cell type composition of the sample and the sequencing coverage are also important factors to consider.

Importantly, our benchmarks also show that our methods increase performance even in small data sets. The use of VMRs over promoters or genomic tiles, for instance, improves the ability to distinguish cell types even in 100-cell data sets, which means that our methods can aide the analysis of such small data sets.

[2] In a few places in the paper (kernel smoothing and detection of variable and differentially methylated regions) the methods use small, fixed size windows, as the basis for analysis - often 1000bp. This size of window can be problematic in BS-Seq data given the uneven distribution of CpGs across the genome, and the potential for this to introduce bias into the results. Would smoothing in windows of fixed numbers of CpGs make more sense?

This is an excellent suggestion. We imagine that using a fixed number of CpG sites instead of a fixed number of base pairs might indeed be more sensible, as this approach would prevent the evaluation of near-empty genomic windows, i.e. windows in genomic regions with little CpG density that might only contain one or two CpG sites. Furthermore, the proposed approach would create smaller windows in CpG-dense areas, which would lead to a more fine-grained evaluation of CpG islands. To put this suggestion to the test, we created a fork of the `scbs` package at https://github.com/LKremer/scbs/tree/adaptive_bandwidth where the sliding window used for VMR detection always comprises a fixed number of CpG sites. Testing this approach on our own multi-omics data set, using 9 CpG sites per window, produced very promising results: We found that VMRs detected with this approach led to a slightly better neighbor score (i.e. better cell type separation), as well as decreased runtime of the VMR detection procedure.

For now, we consider this implementation experimental as its currently lacking unit tests as well as extensive benchmarks on multiple data sets. Furthermore, there are some implementation details that we have not resolved yet: Especially in smaller data sets with low coverage and/or few cells, there might be substantial gaps with zero coverage between individual CpG sites. Does it nonetheless make sense to include adjacent CpG sites into one window, or should there be a maximum distance at which windows are broken into smaller pieces? For now, we thus decided to stick to our tried and tested sliding window approach, but if future benchmarks confirm the superiority of this approach we will update the `scbs` package accordingly.

[3] When selecting the variably methylated regions the method takes the mean of the residuals in a given region. Does this mean that if adjacent regions changed in opposite directions that this would be averaged away and the region discarded as uninteresting?

VMR coordinates are determined by merging overlapping genomic windows above a variance threshold (2% by default). Although the variance is then re-calculated for the entire VMR, this value is not used for a secondary filtering step but only reported in the output file of `scbs scan`.

The rationale behind this is that we also want to report regions where the variance of individual windows briefly drops below the variance threshold - such as the one depicted in Figure 2:

In this example, two of the 2kb windows in the center of the VMR are just barely below the variance threshold (notice the two dots below the dotted red line), but we nonetheless report the entire region because the 2kb windows above the threshold are 2kb wide and are thus able to “bridge the gap”. Similarly, in the scenario described by the reviewer, a VMR with two opposite trends would not be discarded as long as the individual 2kb windows are above the 2% variance threshold. However, our data indicate that such regions are rather rare. A closer look at the variances of all 2kb windows of the largest chromosome, as well as the variances of reported VMRs, shows that the majority of VMRs are above the 2% variance threshold (red line).

As expected due to the two scenarios described above, the region-wide variance of some VMRs is slightly below the 2% window variance threshold. However, these variances are still well

above genome-wide background levels (blue line: average of all genomic windows), suggesting that cancellation of two opposing signals as described by the reviewer is not a common occurrence.

The authors illustrate the effectiveness of their method by showing a PCA of a set of cells separated by their VMR process, compared to a more conventional separation by simple methylation calculation. In the VMR PCA there are two potentially influential steps - the conversion of methylation to residuals, and the imputation of missing values by iterative PCA. It would be useful to know which of these was having the greatest influence, since it's possible that it is the imputation which is causing the clearer separation through the reinforcement of the initial signal, rather than that the initial values are clearer.

It is true that our previous benchmarks did not allow us to distinguish the effects of the choice of dimensionality reduction technique (e.g. iterative PCA) from the choice of DNA methylation measure (average methylation % or shrunken mean of the residuals). To disentangle the influence of these analysis options, we thus refined our benchmarks by testing all possible combinations of the following choices:

- genomic features at which DNA methylation is quantified (100kb tiles, VMRs, promoters or ENCODE candidate cis-regulatory elements (cCREs))
- the measure used to quantify DNA methylation at these features (average methylation % or the shrunken mean of the residuals)
- dimensionality reduction methods (iterative PCA as proposed by us, a software for dimensionality reduction (MOFA+), lightly-imputed PCA on high-coverage features as proposed by Luo et al., or PCA with the missing values set to the column-mean (i.e. zero when using residuals) as suggested by this reviewer in comment [4])

Two of these options, namely the choice of genomic features and the measure of DNA methylation, are distinguished in the revised version of Figure 4:

In Supplementary Figure S1 we furthermore distinguish between four different dimensionality reduction techniques as requested by the reviewer:

Based on these new results, we can now draw more detailed conclusions on the impact of various choices of methods:

- Using the shrunken means of the residuals over average methylation percentages provides the greatest benefit when quantifying promoters or 100kb tiles. The likely reason for this is that these regions are more heterogeneously methylated than VMRs or cCREs: 100kb tiles are very large and

thus each tile will comprise stretches of DNA with high and low methylation. Similarly, many promoter regions (here: TSS±2kb) will comprise a lowly-methylated center region near the TSS, as well as more highly methylated promoter flanks. As the shrunken mean of the residuals is designed to account for this heterogeneity within a region (Fig. 1), it makes sense that we see the greatest benefit of their use when quantifying promoters or large tiles.

2. The use of iterative PCA over other PCA-variations constantly improved the ability to distinguish cell types. The reason for this performance gain is likely that iterative PCA alleviates differences between low-coverage and high-coverage cells, as we show in our reply to the next reviewer question [4].
3. Overall, it is clearly visible that the most important choice is the set of genomic features to be quantified. Our results indicate that both VMRs and ENCODE cCREs are suitable for this task and offer good ability to distinguish cell types, although the use of cCREs is impractical for large data sets due to tremendous increases in RAM and runtime requirements (Supplementary Figure S3).

[4] What would a PCA of the VMR values with missing values still set to 0 look like? This would be a more direct comparison to the global calculation and would demonstrate which step is more important in the improved separation they show.

As mentioned in our reply to the previous question, we expanded our benchmarks by including different strategies for dimensionality reduction, including PCA on a zero-imputed methylation matrix (see Supplementary Figure S1). Note that we labeled this approach “mean-imputed PCA” in our figure, but of course the mean of each genomic feature is 0 in the case of shrunken residuals. As mentioned above, this mean-imputed PCA produces slightly worse results than iterative PCA. The likely reason is that low-quality cells with many imputed zero-values aggregate in the center of PCA space, as is visible in this side-by-side comparison of the two dimensionality reduction approaches (Luo et al. data set):

Although it is clear that iterative PCA remedies technical artefacts introduced by sequencing coverage, our extended benchmarks (Supplementary Figure S1) demonstrate that our performance gains are not primarily driven by the choice of dimensionality reduction. Instead, the most important factor is the choice of genomic intervals, i.e. the use of VMRs over e.g. 100 kb genomic tiles.

[5] In the DMR detection they show a comparison of 130 cells and 58 cells, but am I right in thinking that these cells were split off only after calculating the normalised VMR values from a much larger population of cells? Does the approach still work as effectively if starting from

the 138 cells in the DMR analysis, or do you require the context of the larger dataset to get accurate VMR values to put into the DMR detection?

Yes, the 130 vs 58 cells used for DMR detection were split off from a larger data set (comprising 540 cells) and compared against each other, as is commonly done in scRNA-seq where users might select two cell clusters for differential gene expression testing. To clarify, the DMR detection procedure does not use any VMR coordinates that were detected on the full data set, as DMR coordinates are newly discovered only on the sub-set of cells that the user manually selected. However, as the reviewer rightly points out, the DMR detection uses the shrunken mean of the residuals as a measure of methylation, which in turn uses the smoothed average genomic methylation values for normalization. These smoothed averages were indeed calculated on the full data set and not on the reduced set of 130 + 58 cells. To assess whether this would affect DMR detection, we thus repeated the DMR analysis depicted in Fig. 6 on the 130 + 58 cells only, using smoothed genomic averages calculated on these cells only. The obtained oligodendrocyte- and NSC-DMR sets are largely consistent with those obtained using smoothed averages obtained from the full data set. For comparison, here is Fig. 6 from the manuscript (full data set) and Fig. 6 based on the reduced data set only:

All 540 cells (current manuscript version of the figure):

130 + 58 cells only (please excuse the fact that this figure is not as polished as the version in the manuscript):

While the DMR lists obtained using both approaches are not 100% identical, the results are qualitatively similar and yield largely identical top 5 enriched GO terms.

[6] In the DMR calculation the false discovery rate is estimated by shuffling cell labels and repeating the analysis - which seems to be a reasonable idea for this type of data. However, the FDR estimation is done only after the initially calculated windows are filtered for the top 2% then merged if they are adjacent. The values tested are those from the recalculated t-statistic after merging. I would assume that the approach shown heavily favours regions which are much larger than the initial window size, and that the major contribution to significance is adjacency in the initially filtered top 2% of regions. Does this method work if the FDR values are calculated from the original window data, before merging adjacent windows? If not, then what size of DMR is realistically detectable with this method?

It is correct that we estimate the FDR only after adjacent windows are merged. The reason for this is that we want DMR detection to be flexible, i.e. we want to be able to detect both long and short stretches of differentially methylated DNA. To test whether our DMR detection approach

would also work without merging of adjacent windows, we repeated the DMR detection example from our manuscript but disabled the window-merging functionality (i.e. here we are testing non-overlapping 2 kb genomic bins):

The color code in these two plots is $-\log_{10}$ of the adjusted p-value. As is clearly visible in these two plots, our method detects many more significant DMRs in the real comparison (NSCs vs. oligodendrocytes) compared to the permuted data, even though window-merging was disabled here. Thus, to answer the reviewers question, our method still works even without merging of adjacent windows, but of course this modified version would be unable to report long stretches of differentially methylated DNA.

Reviewer #3:

Remarks to the Author: Kremer et al describe ‘scbs’ – a computational toolset for the analysis of single-cell methylation data. In the field there is certainly a need for a suite of tools to handle single-cell methylation data, and the manuscript details several compelling strategies and frameworks for doing so. The most significant aspect is the identification of variable regions that are then used for subsequent dimensionality reduction, clustering and visualization; followed by tools for DMR calling. The authors apply ‘scbs’ to their own, published, scNMT-seq dataset on brain as well as some applications to a dataset produced by the Ecker Lab, also brain. While there are positives to the work, there are key limitations in the assessment of the tool that need to be performed in order to properly evaluate the tool and its broader utility. Lastly, the name itself is a term that is already used to define an experimental approach” “single-cell bisulfite sequencing” – whereas this reports on a computational tool for that type of data (or enzymatic-converted data, which is not bisulfite). I strongly suggest changing the name of the tool to something informative – e.g. SCMtools for “single-cell methylome tools” – anything that indicates that it is a computational tool and used for analysis of single-cell methylome data.

The name of our tool still stems from the times of our very first internal prototype, and admittedly we are not quite happy with it ourselves. Although the tool is already released to the public by now, we are planning to rename it before publication, i.e. before it reaches a wider audience.

Major Comments:

The authors perform almost all of their evaluation solely on their own dataset with very little analysis of the Luo et al 2017 dataset. Furthermore, their only comparison of methodology is by using 100 kbp tiling windows for CpG methylation levels, which does not seem all that appropriate. Using relevant windows – e.g. all annotated promoters and enhancers like geneHancer or something like that is a much more reasonable strategy. Even just all promoter regions such as TSS +/- 2 kbp or something.

We thank the reviewer for this helpful remark, which prompted us to greatly expand the scope of our benchmarks. To address the concern that our evaluation is mostly based on our own data, we incorporated two additional publicly available data sets into our benchmarks: a single-cell data set comprising human colorectal cancer samples (Bian et al., <https://doi.org/10.1126/science.aao3791>) and the mouse embryo data set kindly suggested by the reviewer (Argelaguet et al., <https://doi.org/10.1038/s41586-019-1825-8>). As suggested by the reviewer, we also included two other sets of genomic intervals into our benchmarks, namely annotated regulatory regions (ENCODE candidate cis-regulatory elements, cCREs) and promoter regions (TSS ± 2 kb). Our benchmarks now comprise four distinct data sets, four sets

of genomic intervals, and four strategies for dimensionality reduction (Supplementary Figure S1, see below).

This comprehensive benchmark demonstrates that our VMR-detection approach is robust and yields good results on diverse data sets. cCREs also produced good results in most methods combinations. However, only VMRs but not cCREs performed well when MOFA+ (Argelaguet *et al.* 2020) was used for dimensionality reduction.

It is worth noting that the set of cCREs comprises ca. 340,000 genomic intervals while we detected ca. 50,000 to 60,000 VMRs. Thus, the use of cCREs proved challenging in practice, as the large number of cCREs results in very large methylation matrices. Analyzing the full Luo *et al.* data set, for instance, took roughly 2 hours and 96 GB RAM when using cCREs as features, while the same analysis using VMRs merely took 5 minutes (+ 11 min for VMR detection) and 15 GB RAM. We attempted to remedy this problem by discarding low-coverage cCREs. While this reduced runtime and RAM usage, this approach also reduced performance, suggesting that VMRs are more information-rich than cCREs (Supplementary Figure S3, see below).

They do not compare cluster resolution with CH methylation windows which is generally used for neuron subtype clustering (as often CpG methylation will be the same between two subtypes of the same class, eg excitatory, but the CH levels will be different). Does the tool work with CH methylation? How does the variable window approach for CpG methylation look compared to 100 kbp tiling CH methylation (where 100 kbp tiles make sense due to the structure of the mark in neurons).

As CH methylation is commonly used to distinguish neuronal subtypes, we agree that it is worthwhile to assess whether our tool also works with this type of data. To test this, we re-analyzed the Luo et al. data set using three different strategies:

- the classic approach used by Luo et al.: averaging CH methylation in 100 kb genomic tiles
- applying our approach on CH methylation data: using the CH data for VMR detection, quantifying CH methylation in these intervals with the shrunken mean of the residuals
- applying our approach on CpG methylation data, as was also done in our benchmarks.

Due to the lack of ground truth cell type labels that are independent of the CH/CpG data, we could not quantify cell type separation as done in our benchmarks. Nonetheless, visual inspection

of the three resulting UMAPs clearly showed that all three approaches were able to separate the neuronal subtypes which Luo et al. annotated based on CH methylation in 100 kb genomic tiles (Supplementary Figure S4A, see below). This analysis demonstrates that our tool also works with CH methylation data.

These analyses should be carried out on both their own dataset, the Luo et al dataset, but also on datasets from other tissue types. For example:

Argelaguet R, Clark SJ, Mohammed H, Stapel LC, Krueger C, Kapourani CA, Imaz-Rosshandler I, Lohoff T, Xiang Y, Hanna CW, Smallwood S, Ibarra-Soria X, Buettner F, Sanguinetti G, Xie W, Krueger F, Göttgens B, Rugg-Gunn PJ, Kelsey G, Dean W, Nichols J, Stegle O, Marioni JC, Reik W. Multi-omics profiling of mouse gastrulation at single-cell resolution. *Nature*. 2019 Dec;576(7787):487-491. doi: 10.1038/s41586-019-1825-8. Epub 2019 Dec 11.

Chatterton, Z., Lamichhane, P., Ahmadi Rastegar, D. et al. Single-cell DNA methylation sequencing by combinatorial indexing and enzymatic DNA methylation conversion. *Cell Biosci* 13, 2 (2023). <https://doi.org/10.1186/s13578-022-00938-9> (Enzymatic conversion workflow)

We agree that analyzing further data sets would strengthen the manuscript, which is why we included two additional data sets (four data sets in total) in our benchmarks. As briefly mentioned in our previous reply, benchmarking the performance of different methods requires ground truth cell labels, for instance cell type labels derived from scRNA-seq data of the same cells, to assess how well these labeled groups of cells can be separated by a given method. Unfortunately, most public single-cell methylation data sets only come with cell type labels that were annotated based on the methylation data itself, which makes them unsuitable for benchmarking. Finding suitable data set with good ground truth cell labels is challenging, but

fortunately one of the data sets proposed by the reviewer (Argelaguet et al.) is multi-omic and contains three sets of cell type labels which we were able to use for benchmarking: germ layer, cell lineage and developmental stage of the embryo. As mentioned above, we also found a human colorectal cancer data set with cell labels independent from the methylation data (Bian et al.), which we also included in our benchmarks.

Furthermore, the tool is demonstrated on relatively small datasets on the order of ~3000 cells; however, single-cell methylation datasets are increasing in size – the Luo et al dataset used is quite old at this point, with much larger datasets produced by the Ecker and Luo labs, with hundreds-of-thousands of cells. It is likely that such datasets will become more common and any tool that will reach wide adoption will have to be able to handle large datasets. At least one of these should be used as an example with compute times reported – eg:

Liu, H., Zhou, J., Tian, W. et al. DNA methylation atlas of the mouse brain at single-cell resolution. *Nature* 598, 120–128 (2021). <https://doi.org/10.1038/s41586-020-03182-8> (~100k cells) Or the newer dataset that is ~300-400k cells; however, the 100k cell dataset should be sufficient for demonstration purposes. Again comparing CH methylation 100 kbp windows (standard for neurons) vs variable windows vs promoters & enhancers.

It is true that single-cell data sets, even those which quantify DNA methylation, are constantly increasing in size. Today, such single-cell methylation atlases are still rare and pose a tremendous data analysis challenge that likely takes months to complete — the raw data of the suggested 100k cell data set comprises 20 terabytes of data. Nonetheless, we agree that bigger data sets will become more common in the future, which is why we stress-tested our tool on the 100k cell data set proposed by the reviewer. During the course of this analysis, we identified a technical issue in our code that had so far prevented the analysis of huge data sets. We thank the reviewer for bringing this to our attention. After fixing this issue, we used our default workflow on the 100k cell data set from Liu et al. (2021), and successfully separated the cell types annotated by Liu et al, as illustrated in Supplementary Figure S4B:

B Liu *et al.* data set (~100,000 cells)
CpG, shrunken mean of residuals in VMRs

As expected from the size of the processed data, this analysis had to be performed on a compute cluster with multiple nodes of 256 GB RAM and 48 CPUs each. Since we processed chromosomes in parallel on different nodes, our analysis took less than two days. For labs with access to just one computer of this size, the same analysis would take approximately one week to complete. This seems long, but it is just a fraction of the time required to map single-cell bisulfite-sequencing data for 100k cells (which would also necessitate the use of a large compute cluster). Of note, most of this time (152 hours) is spent by the prepare step, which

decompresses and reads the raw data which we obtained from Liu et al. (~20 terabytes for this data set). Fortunately, this step is only performed once per data set, and all other follow-up steps including VMR detection completed within one day. Thus, we believe that scbs is suitable for the analysis of large data sets, given the same computational resources that would also be required to map the bisulfite-converted reads within a reasonable time frame.

Decision Letter, first revision:

Our ref: NMETH-A52707A

18th Mar 2024

Dear Dr. Anders,

Thank you for submitting your revised manuscript "Analyzing single-cell bisulfite sequencing data with "scbs"" (NMETH-A52707A). It has now been seen by the original referees and their comments are below. The reviewers find that the paper has improved in revision, and therefore we'll be happy in principle to publish it in Nature Methods, pending minor revisions to satisfy the referees' final requests and to comply with our editorial and formatting guidelines.

TRANSPARENT PEER REVIEW

Please note: we allow redactions to authors' rebuttal and reviewer comments in the interest of confidentiality. If you are concerned about the release of confidential data, please let us know specifically what information you would like to have removed. Please note that we cannot incorporate redactions for any other reasons. Reviewer names will be published in the peer review files if the reviewer signed the comments to authors, or if reviewers explicitly agree to release their name. For more information, please refer to our FAQ page.

ORCID

IMPORTANT: Non-corresponding authors DO NOT have to link their ORCIDs but are encouraged to do so. Please note that it will not be possible to add/modify ORCIDs at proof. Thus, please let your co-authors know that if they wish to have their ORCID added to the paper they must follow the procedure

described in the following link prior to acceptance:

Sincerely,
Lei

Lei Tang, Ph.D.
Senior Editor
Nature Methods

Reviewer #1 (Remarks to the Author):

The authors have adequately addressed all my comments in their responses. Please publish this important paper. Congratulations!

Reviewer #1 (Remarks on code availability):

I can install the package from pip successfully.

Reviewer #2 (Remarks to the Author):

The revision and comments supplied by the authors along with additional figures and analyses have addressed the concerns I raised in the initial review. There are areas where caution will need to be taken when interpreting results from this method, but the main steps in the analysis appear to be both based on sound theory and robust in their execution.

Personally I would appreciate the option to build windows from fixed numbers of CpG to make it into the main release, rather than being in a branch at the moment, but this isn't something which is critical enough to be a condition of publication.

Reviewer #2 (Remarks on code availability):

I was able to download and run the code, but I have not done an in-depth review of the actual coding. The codebase they supplied came with a well constructed test-suite and I have no reason to doubt the implementation claimed in the manuscript.

Reviewer #3 (Remarks to the Author):

The expanded analyses and clarifications provided by the authors for each of my comments satisfies my concerns.

Reviewer #3 (Remarks on code availability):

The code was able to be used successfully on datasets beyond those detailed in the manuscript.

Author Rebuttal, first revision:

Reviewer #1:

Remarks to the Author:

The authors have adequately addressed all my comments in their responses. Please publish this important paper. Congratulations!

We thank the Reviewer for these kind words and for their constructive and fair assessment of our work.

Reviewer #2:

Remarks to the Author:

The revision and comments supplied by the authors along with additional figures and analyses have addressed the concerns I raised in the initial review. There are areas where caution will need to be taken when interpreting results from this method, but the main steps in the analysis appear to be both based on sound theory and robust in their execution.

Personally I would appreciate the option to build windows from fixed numbers of CpG to make it into the main release, rather than being in a branch at the moment, but this isn't something which is critical enough to be a condition of publication.

We thank the Reviewer for their helpful feedback on our manuscript and for their fair assessment. We will add an option to build windows from fixed numbers of CpG sites in a future release of the

tool.

Reviewer #3:

Remarks to the Author:

The expanded analyses and clarifications provided by the authors for each of my comments satisfies my concerns.

We thank the Reviewer for their fair evaluation of our study.

Final Decision Letter:

11th Jun 2024

Dear Professor Anders,

I am pleased to inform you that your Article, "Analyzing single-cell bisulfite sequencing data with 'MethSCAN'", has now been accepted for publication in Nature Methods. The received and accepted dates will be 26th May 2023 and 11th Jun 2024. This note is intended to let you know what to expect from us over the next month or so, and to let you know where to address any further questions.

Over the next few weeks, your paper will be copyedited to ensure that it conforms to Nature Methods style. Once your paper is typeset, you will receive an email with a link to choose the appropriate publishing options for your paper and our Author Services team will be in touch regarding any additional information that may be required. It is extremely important that you let us know now whether you will be difficult to contact over the next month. If this is the case, we ask that you send us the contact information (email, phone and fax) of someone who will be able to check the proofs and deal with any last-minute problems.

Please note that Nature Methods is a Transformative Journal (TJ). Authors may publish their research with us through the traditional subscription access route or make their paper immediately open access through payment of an article-processing charge (APC). Authors will not be required to make a final decision about access to their article until it has been accepted. Find out more about Transformative Journals

Authors may need to take specific actions to achieve compliance with funder and institutional open access mandates. If your research is supported by a funder that requires immediate open access (e.g. according to Plan S principles) then you should select the gold OA route, and we will direct you to the compliant route where possible. For authors selecting the subscription publication route, the journal's standard licensing terms will need to be accepted, including self-archiving policies. Those licensing terms will supersede any other terms that the author or any third party may assert apply to any version of the manuscript.

Best regards,
Lei

Lei Tang, Ph.D.
Senior Editor
Nature Methods